# The concerted roles of FANCM and Rad52 in the protection of common fragile sites

Hailong Wang [1], Shibo Li[2], Joshua Oaks[2], Jianping Ren[1], Lei Li[3] & Xiaohua Wu[2]

Common fragile sites (CFSs) are prone to chromosomal breakage and are hotspots for chromosomal rearrangements in cancer cells. We uncovered a novel function of Fanconi anemia (FA) protein FANCM in the protection of CFSs that is independent of the FA core complex and the FANCI–FANCD2 complex. FANCM, along with its binding partners FAAP24 and MHF1/2, is recruited to CFS-derived structure-prone AT-rich sequences, where it suppresses DNA double-strand break (DSB) formation and mitotic recombination in a manner dependent on FANCM translocase activity. Interestingly, we also identified an indispensable function of Rad52 in the repair of DSBs at CFS-derived AT-rich sequences, despite its non-essential function in general homologous recombination (HR) in mammalian cells. Suppression of Rad52 expression in combination with FANCM knockout drastically reduces cell and tumor growth, suggesting a synthetic lethality interaction between these two genes, which offers a potential targeted treatment strategy for FANCM-deficient tumors with Rad52 inhibition.

[1] Beijing Key Laboratory of DNA Damage Response and College of Life Science, Capital Normal University, Beijing 100048, China. [2] Department of Molecular Medicine, The Scripps Research Institute, La Jolla, CA 92037, USA. [3] Department of Genetics, The University of Texas M.D. Anderson Cancer Center, Houston, TX 77030, USA. These authors contributed equally: Hailong Wang, Shibo Li. Correspondence and requests for materials should be addressed to X.W. (email: xiaohwu@scripps.edu)

Common fragile sites (CFSs) are large chromosomal regions where gaps and breaks are recurrently generated upon replicative stress[1]. They are preferentially unstable during the early stages of cancer development and often associated with chromosomal rearrangement sites in tumors[2–4]. Compelling evidence suggests that perturbation of DNA replication at these regions is a major cause for CFS instability[5]. Aberrant oncogene expression promotes CFS breakage (often called CFS expression)[6], likely due to oncogene-induced replication stress[7,8]. It is believed that CFS instability is one driving force for tumorigenesis.

CFSs are enriched with interrupted AT-dinucleotide repeats (AT-rich), which are predicted to form DNA secondary structures[9]. Such AT-rich sequences derived from FRA16D cause replication fork stalling, and induce double-strand break (DSB) formation and mitotic recombination[10,11]. DNA combing analysis also demonstrated that fork arrest at the endogenous FRA16C site is preferentially close to the AT-rich sequences[12]. Thus, forming DNA secondary structures at CFSs is an important factor to induce fork stalling and CFS destabilization.

Cytogenetic studies have revealed that chromosomal breakpoints in Fanconi anemia (FA) patients often colocalize with CFSs[13]. Consistently, FA proteins play important roles in CFS protection[14]. FA is a genetically heterogenous disorder characterized by severe genome instability, extreme sensitivity to interstrand crosslinking (ICL) agents, developmental abnormalities, bone marrow failure, and cancer predisposition[15,16]. Upon DNA damage, the FA core complex (composed of FANCA, FANCB, FANCC, FANCE, FANCF, FANCG, FANCL, and FANCM along with their association proteins) is required for monoubiquitinating the FANCD2 and FANCI heterodimeric complex (ID2), which marks the activation of the FA pathway. Downstream FA proteins including FANCD1/BRCA2, FANCJ/BRIP1, FANCN/PALB2, and FANCO/RAD51 are important for homologous recombination (HR)-mediated DSB repair.

FANCM is a component of the FA core complex and it also forms tight complexes with its binding partners FAAP24 and MHF1/2 (MHF)[17,18]. It contains an N-terminal DEAH helicase domain and exhibits an ATP-dependent DNA-remodeling translocase activity[19]. The localization of the FA core complex to chromatin and monoubiquitination of ID2 complex require FANCM but not its translocase activity[20–22]. In vitro, FANCM binds specifically to model replication forks and Holliday junctions and promotes fork reversal and migration of junction points in an ATPase-dependent manner[23,24]. Biochemical studies have demonstrated that both MHF and FAAP24 stimulate DNA binding by FANCM, and MHF promotes the fork remodeling activity of FANCM[17,18,25]. In this study, we have identified a new role of FANCM in the maintenance of CFS stability, that is independent of the previously described function of the FA core and ID2 complexes in CFS protection[14,26], but requires its translocase activity and binding partners FAAP24 and MHF.

HR plays an important role in CFS protection[27], and it has been shown that the AT-rich sequence Flex1 derived from FRA16D induces HR-mediated mitotic recombination[10]. In mammalian cells, Rad51, BRCA1, and BRCA2 are required for HR[28]. However, despite an essential function of Rad52 for HR in yeast, Rad52 is not required for HR in mammalian cells[29,30]. Knockout (KO) of the *Rad52* gene in mice has almost no phenotype in recombination and repair; this is different from *Rad51* KO, which shows early embryonic lethality[31–33]. In this study, we found a novel function of Rad52 for repairing DSBs accumulated at the AT-rich sequences present in CFSs when FANCM is deficient. Combined inactivation of FANCM and Rad52 leads to a strong cell proliferation defect, suggesting a synthetic lethality interaction between these two genes.

## Results

### FANCM suppresses Flex1-induced mitotic recombination in a manner independent of the FA core complex.
We showed that the AT-rich sequence Flex1 derived from FRA16D is genetically unstable, and induces HR-mediated mitotic recombination, as revealed by the EGFP-based HR reporter, HR-Flex (containing Flex1) using HR-Luc reporter (containing a luciferase fragment) as a control[10] (Fig. 1a and Supplementary Fig. 1a). We further showed that the AT-rich sequences 16C-AT1 and 16C-AT3 derived from FRA16C, and 3B-AT derived from FRA3B also induce mitotic recombination in a similar manner as Flex1 (Supplementary Fig. 1b and 1c). Thus, it is a common feature that AT-rich and structure-forming DNA sequences derived from CFSs are unstable and induce mitotic recombination.

To further study the mechanisms of CFS protection, we examined mitotic recombination using the HR-Flex reporter after inactivation of different repair proteins. Depletion of Rad51, Mre11, CtIP, BRCA1, or BRCA2 by shRNAs leads to a reduction of Flex1-induced mitotic recombination (Supplementary Fig. 2a and 2b), consistent with the role of these proteins in HR[28]. Interestingly, suppression of FANCM expression leads to a substantial increase of spontaneous mitotic recombination at Flex1 in U2OS cells with HR-Luc as a control (Fig. 1b and Supplementary Fig. 2c). Similar increase of mitotic recombination at Flex1 was observed in other cell lines including Hela, MCF7, and T98G when FANCM expression is suppressed by shRNAs (Fig. 1c and Supplementary Fig. 3a). In addition, mitotic recombination at Flex1 and the AT-rich sequence (3B/AT) derived from FRA3B is substantially higher in HCT116 FANCM KO cells compared to HCT116 wild-type cells (Fig. 1d and Supplementary Fig. 3b).

Different from FANCM, depletion of the FA core complex protein FANCA and FANCD2 results in a much weaker effect (Fig. 1b and Supplementary Fig. 2c). Furthermore, the FANCM-MM1 mutant, which is defective in FANCM binding to the FA core complex[34], suppresses Flex1-induced mitotic recombination to a similar level as FANCM-WT (Fig. 1e). Thus, the significant role of FANCM in suppression of Flex1-induced mitotic recombination is largely independent of the FA core and ID2 complexes. We further showed that when a DSB is generated adjacent to Flex1 or Luc by the endonuclease I-SceI, depletion of FANCM, FANCA, or FANCD2 similarly leads to a detectable but mild defect in HR (Fig. 1f and Supplementary Fig. 4), in agreement with the previous findings that the FA pathway is involved but not essential for HR[35,36]. These data suggest that FANCM possesses a unique activity in the suppression of Flex1-induced mitotic recombination prior to the DSB formation and HR-mediated repair at Flex1.

### The translocase activity of FANCM is required for suppressing Flex1-induced mitotic recombination.
FANCM exhibits translocase activity, which is important for its fork regression function[19]. We reasoned that upon ssDNA exposure during DNA replication, Flex1 could form secondary structures causing replication stalling and DSB formation. FANCM may promote fork reversal to remove such DNA secondary structures, thus preventing DSB formation at Flex1 (see Discussion). In support of this model, while FANCM-WT suppresses hyper-mitotic recombination at Flex1, FANCM translocase mutant K117R fails to do so (Fig. 1g).

DSBs are accumulated at Flex1 after HU treatment as revealed by ChIP analysis of γH2AX at Flex1 (Supplementary Fig. 5a). Depletion of FANCM by shRNAs or expression of FANCM translocase mutant (FANCM-K117R) with endogenous FANCM silenced by shRNAs leads to substantial increase of γH2AX at

Flex1 upon HU treatment (Fig. 2a, left and middle, and Supplementary Fig. 5b). This is in accordance with the model that the translocase activity of FANCM is important for removing DNA secondary structures at Flex1 to prevent DSB formation.

**FAAP24 and MHF1 are important for FANCM to be recruited to Flex1 to suppress Flex1-induced mitotic recombination.** As revealed by ChIP analysis, FANCM is recruited to Flex1 in the HR-Flex reporter that is stably integrated into the genome, and

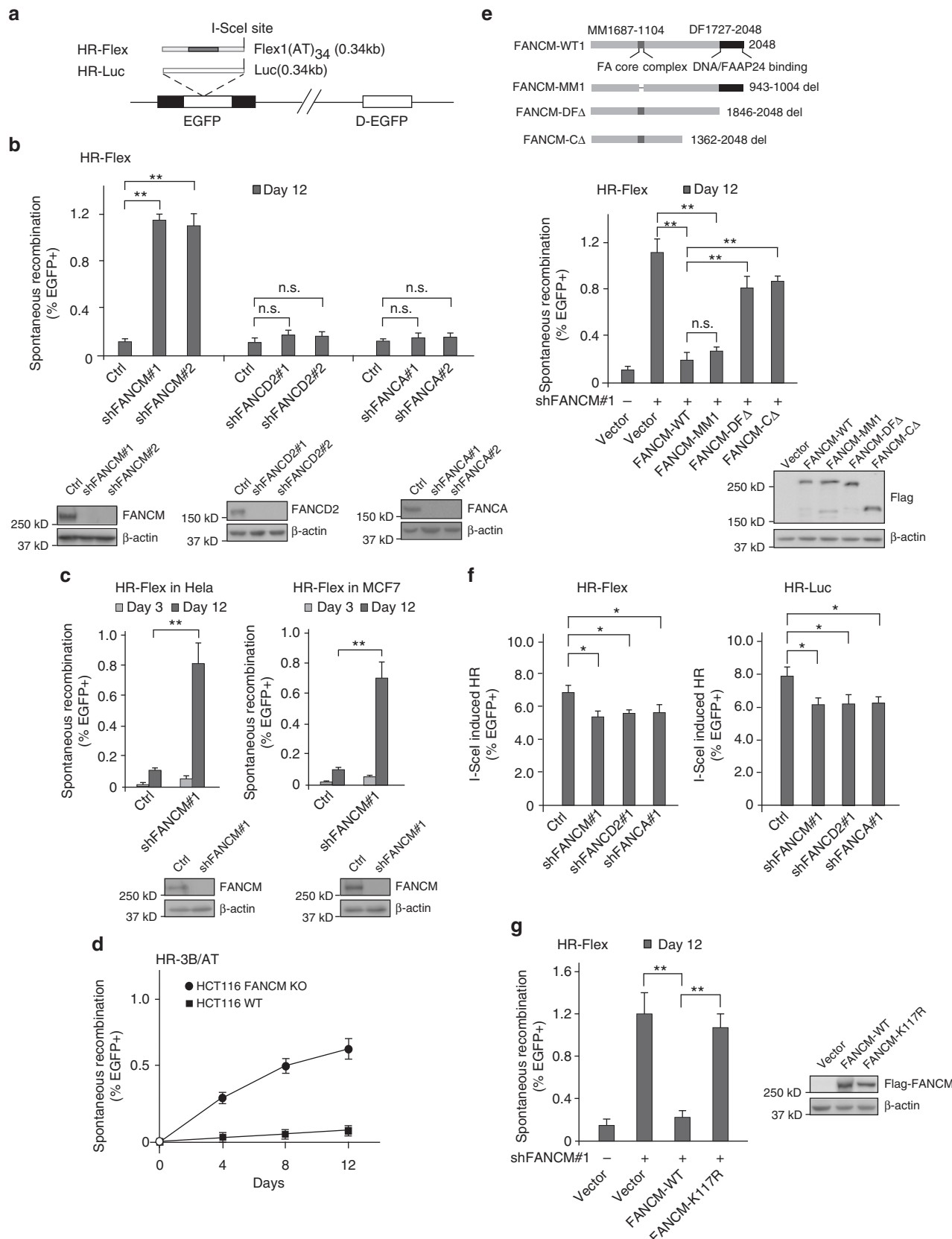

such recruitment is further increased after aphidicolin (APH) treatment (Fig. 2b). FANCM binding partners FAAP24 and MHF1 are also recruited to Flex1 (Supplementary Fig. 5c). Depletion of FAAP24 or MHF1 by shRNAs significantly impairs the recruitment of FANCM to Flex1 (Fig. 2c and Supplementary Fig. 5d), and results in DSB accumulation and hyper-mitotic recombination at Flex1 (Fig. 2a, right, 2d, and Supplementary Fig. 6). While the translocase mutant FANCM-K117R is recruited normally, the C-terminal deletion mutants FANCM-DFΔ (Δ1846–2048) and FANCM-CΔ (Δ1362–2048), impaired in both DNA and FAAP24 binding[34], are defective in being recruited to Flex1 (Fig. 2e and Supplementary Fig. 5e). The FANCM-DFΔ and FANCM-CΔ mutants also fail to suppress mitotic recombination at Flex1 (Fig. 1e). These data suggest that FAAP24 and MHF assist FANCM to be recruited to Flex1, which is important for the suppression of Flex1-induced mitotic recombination.

Further analysis showed that the DNA binding-defective mutant FAAP24-DB (K171A/K173A)[37], but not the FANCM binding mutant FAAP24-MB (V198A/V199A/G200A)[38], is impaired in recruitment to Flex1 (Fig. 2f, left and Supplementary Fig. 5f). These data suggest that the binding of FAAP24 to DNA, but not the interaction of FAAP24 with FANCM is important for FAAP24 to be recruited to Flex1. However, both mutants are defective for recruiting FANCM to Flex1 and for suppressing Flex1-induced mitotic recombination (Fig. 2f, right, 2g, and Supplementary Fig. 5g). Thus, the interaction of FAAP24 with FANCM is required for FANCM to be recruited to Flex1 and both DNA binding and FANCM binding activities of FAAP24 are important for suppressing Flex1-induced mitotic recombination.

**FANCM is important for maintaining the genome stability at Flex1 and protecting CFSs.** Flex1 induces plasmid instability in mammalian cells[10]. We further showed that when FANCM, FAAP24, or MHF1 is depleted by shRNAs, Flex1-containing plasmids become even more unstable (Fig. 3a), likely due to DSB formation at Flex1 when the FANCM pathway is impaired (Fig. 2a). Importantly, FANCM translocase activity is important for preventing Flex1-induced plasmid instability (Fig. 3b and Supplementary Fig. 7).

To examine whether FANCM is important for CFS stability, we spread metaphase chromosomes and found increased chromosomal breakages per cell when FANCM is depleted by shRNAs, which is further elevated following APH treatment under the condition that induces CFS expression (Fig. 3c). To more specifically look at CFSs, we performed fluorescence in situ hybridization (FISH) analysis at FRA16D and FRA3B on metaphase spreads. We showed that FRA16D and FRA3B expression is significantly increased when FANCM is depleted (Fig. 3d). ChIP analysis of γH2AX at endogenous FRA3B locus showed that DSBs are accumulated at the vicinity of AT-rich sequences in FRA3B (Fig. 3e). We further showed that FRA16D and FRA3B expression is also increased when FANCM

translocase activity is deficient (Fig. 3f). These data suggest that FANCM and its translocase activity are important for CFS protection to prevent DSB formation.

FRA16D expression is also increased in the FANCM-MM1 mutant defective in binding to the FA core complex, but to a less extent compared to the translocase mutant FANCM-K117R (Supplementary Fig. 8). Since mitotic recombination at Flex1 is not significantly increased in the FANCM-MM1 mutant (Fig. 1e), increased CFS expression in this mutant is not due to a defect of FANCM in replication fork remodeling, but likely in coordination with the FA core complex to protect Flex1. This is consistent with the observation that the FA core complex is important for maintaining CFS stability[14] but not for the suppression of Flex1-induced mitotic recombination (Fig. 1b). These data also suggest that FANCM protects CFSs mainly through utilizing its translocase activity to preserve stability of AT-rich sequences at CFSs, but the interaction of FANCM with the FA core complex also contributes to CFS protection.

**FANCM protects CFSs upon oncogene expression.** Replication stress is induced upon oncogene expression[7,8]. Consistently, we observed that H-Ras-V12 overexpression induces ATR activation and DSB formation as revealed by phosphorylation of Chk1 and RPA, and phosphorylation of H2AX, respectively (Supplementary Fig. 9). In accordance with the previous study[6], overexpression of H-Ras-V12 induces FRA16D expression (Fig. 4a). We further showed that FANCM and its translocase activity are required for suppression of Ras-induced CFS expression (Fig. 4b, c).

Ras overexpression also induces mitotic recombination at Flex1 (Fig. 4d), likely due to replication stress caused by oncogene expression leading to DSB formation. Indeed, ChIP analysis of γH2AX shows increased DSB formation at Flex1 upon Ras overexpression (Fig. 4e). Loss of FANCM or FANCM translocase activity, as well as suppression of FAAP24 or MHF1 expression, further increases Ras-induced mitotic recombination at Flex1 (Fig. 4f, g). These data suggest that FANCM in complex with FAAP24 and MHF, and the translocase activity of FANCM are important for suppressing oncogene-induced CFS instability, which often occurs at the early stage of cancer development.

**Rad52 is important for HR at Flex1.** Loss of FANCM significantly increases spontaneous mitotic recombination as revealed by the HR-Flex reporter (Fig. 1b), suggesting that HR is used to repair DSBs generated at Flex1 upon loss of the protection mechanism by FANCM. Indeed, inactivation of Rad51 reduces green cell accumulation in FANCM-knockdown HR-Flex reporter cell line (Fig. 5a and Supplementary Fig. 10a). Surprisingly, similar effect was also observed when Rad52 is depleted, although Rad52 is not required for general HR in mammalian cells[33,39]. Similarly, inactivation of Rad51 or Rad52 both suppresses HU-induced mitotic recombination at Flex1 (Fig. 5b and Supplementary Fig. 10b). ChIP analysis demonstrated that both Rad51 and Rad52 are recruited to Flex1 (Fig. 5c). These data suggest that

**Fig. 1** FANCM is important for suppressing Flex1-induced mitotic recombination independent of the FA core complex and ID2 complex. **a** Schematic drawing of the EGFP-based HR-Flex and HR-Luc reporters as previously described[10]. Luc: luciferase fragment; D-EGFP: donor EGFP. **b**, **c** shRNAs for indicated proteins or a control vector (Ctrl) were expressed in U2OS (**b**) or Hela and MCF7 (**c**) cells containing the HR-Flex reporter, and assayed for spontaneous mitotic recombination after culturing for 3 or 12 days. The expression of indicated protein was examined by western blot analysis. **d** Spontaneous recombination was examined in HCT116 wild-type (WT) and HCT116 FANCM KO cells carrying the HR-3B/AT (AT-rich sequences derived from FRA3B) reporter after culturing pre-sorted non-green cells for indicated days. **e**, **g** Spontaneous recombination was assayed in U2OS (HR-Flex) cells expressing Flag-FANCM WT or indicated mutants 12 days after endogenous FANCM was silenced by shRNA. The expression of Flag-FANCM WT and indicated mutants is shown by western blot analysis. **f** The cell lines described in **b** were infected with I-SceI retroviruses and assayed for HR 5 days later. In all experiments, error bars represent standard deviation (SD) of at least three independent experiments. The P value is indicated as *P < 0.05 and **P < 0.01. n.s. not significant. Western blot analysis was performed using β-actin or Ku70 as a loading control

Rad52 is involved in HR to repair DSBs generated at Flex1. Indeed, inactivation of Rad52 in FANCM-deficient cells further increases DSB formation at Flex1 as revealed by ChIP analysis of γH2AX (Fig. 5d).

To further illustrate the mechanism underlying the role of Rad52 in mitotic recombination at Flex1, we used I-SceI to create a DSB in the HR reporters with or without Flex1 insertion (Fig. 5e, left and middle). Suppression of Rad52 expression has no

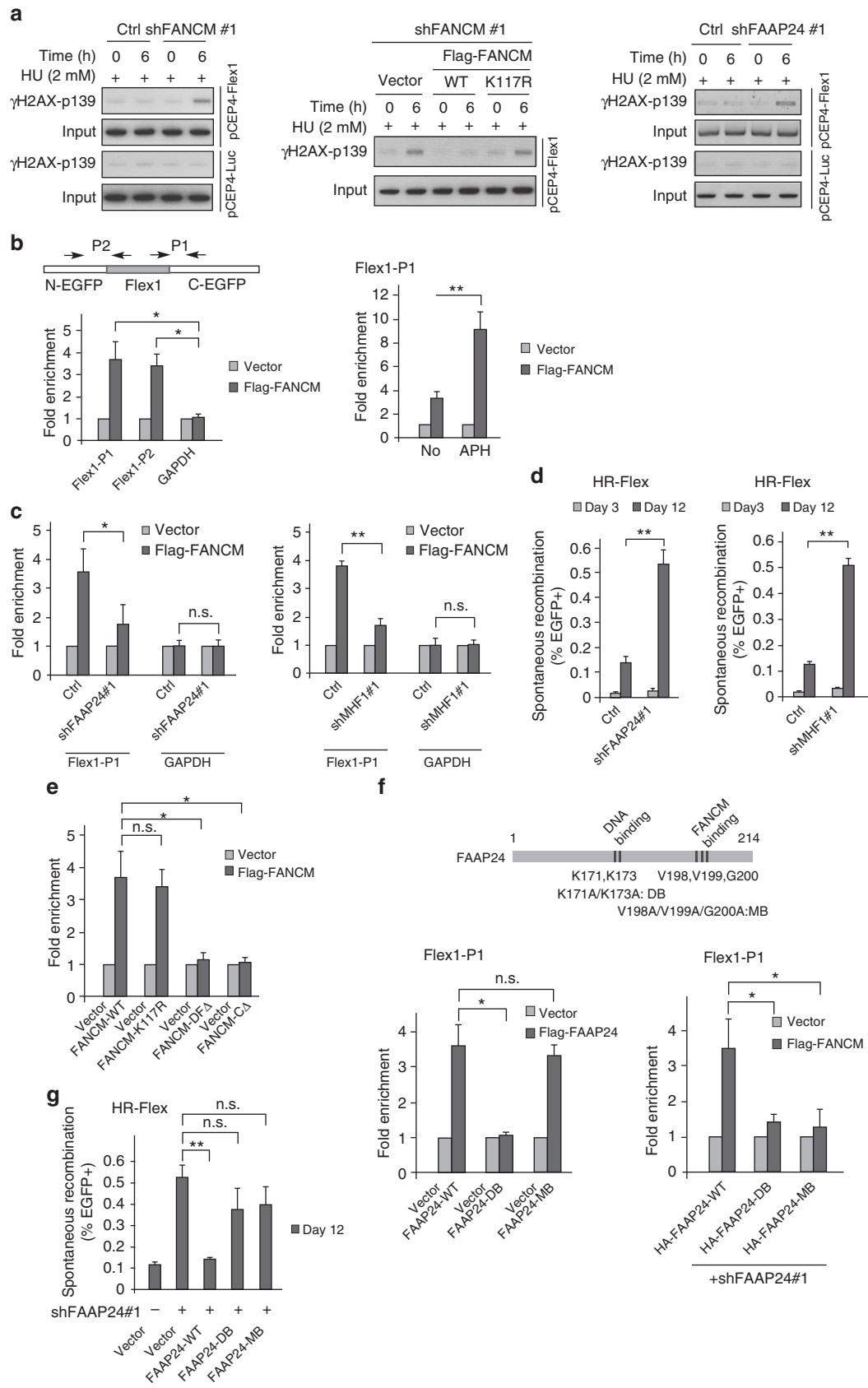

obvious effect when using the general HR reporter without Flex1, but leads to a significant reduction of HR when HR-Flex was used. We reasoned that the presence of Flex1 at the DSB end as a non-homologous sequence to the template, which also forms DNA secondary structure, may block the end and inhibit strand invasion or second-end capture, and consequently Rad52 becomes important for initiating and/or completing HR (Supplementary Fig. 11 and Discussion). When we used the HR-Luc reporter which contains a luciferase sequence (similar size to Flex1) at the side of the I-SceI cleavage site, Rad52 is also required for HR (Fig. 5e, right). These data suggest that when one DSB end is blocked by a non-homologous sequence to the template, Rad52 becomes essential for HR in mammalian cells.

Commonly, sister chromatids serve as homologous templates for HR-mediated DSB repair, and non-homologous sequences are not present in the sister chromatid templates. We hypothesize that when DSB ends are blocked by DNA secondary structures even without non-homologous sequences present, Rad52 may also be required for promoting HR from a blocked end (see Discussion). To test this hypothesis, we designed DSB repair substrates (HR-Flex/D-Flex and HR-Luc/D-Luc, Fig. 5f, top) with Flex1 or Luc also inserted in the EGFP donor templates. Due to the presence of Flex1 and Luc in the donor templates, HR would not produce green cells and PCR analysis was used to score the repair events. After I-SceI cleavage, end resection would occur leading to single-stranded DNA (ssDNA) accumulation at DSB ends and Flex1 but not Luc would form secondary structures. Genomic DNA was purified from the reporter cell lines after I-SceI expression and digested with I-SceI to remove the parental EGFP recipient cassettes (uncut by I-SceI or perfectly religated after I-SceI cleavage in cells), followed by PCR with primers specifically for the parental/repaired EGFP recipient cassettes. The donor templates are marked by BamHI and EcoRI sites, which are absent in the parental cassettes. If HR is used, BamHI and EcoRI sites would be transferred to the HR repair products to replace the I-SceI site, while end joining products would not contain BamHI and EcoRI sites. Thus, the ratio of BamHI and EcoRI cleavable and non-cleavable PCR products (Fig. 5f, right box) would indicate the ratio of the repair products by HR or by imperfect end joining. Interestingly, the percentage of using HR in U2OS (HR-Flex/D-Flex) cells is substantially reduced after Rad52 depletion, but such reduction was not observed in U2OS (HR-Luc/D-Luc) cells (Fig. 5f, bottom). These data suggest that DNA secondary structures formed at Flex1 after end resection would block DSB ends and even when the DSB ends contain perfect homology to the donor templates, Rad52 is required to promote HR-mediated repair from blocked DSB ends.

**Loss of Rad52 and FANCM leads to a strong proliferation defect and inactivation of Rad52 suppresses the growth of FANCM-deficient tumors.** Neither Rad52 nor FANCM is an essential gene in mice[33,40], and inactivation of Rad52 or FANCM does not significantly influence cell growth (Fig. 6a). However, when Rad52 is depleted by shRNAs in FANCM KO cells, a strong proliferation defect is observed, suggesting a synthetic lethality interaction between these two genes. Inhibition of Rad52 in FANCM KO cells expressing FANCM translocase mutant FANCM-K117R also results in a strong cell growth defect, which is in sharp contrast to cells expressing the FANCM wild-type allele (Fig. 6b). These results support the model that Rad52 and FANCM play concerted roles in protecting structure-prone DNA sequences at CFSs, and absence of both would cause cell death.

Inactivation or downregulation of FANCM has been observed in breast cancer and other tumors[41–43]. To test whether depletion of Rad52 inhibits FANCM-deficient tumor growth, we monitored tumor formation in a mouse xenograft model. While the sizes of tumors derived from FANCM KO cells and Rad52 knockdown cells are initially a bit smaller than the control, they grow at similar rate as that from wild-type cells at later time points (Fig. 6c). In a stark contrast, depletion of Rad52 in FANCM KO cells completely suppresses tumor growth. Thus, inhibition of Rad52 could offer a new targeted treatment strategy for FANCM-deficient tumors.

## Discussion

CFSs are extended over hundreds of kilobases and their instability is induced by multiple mechanisms such as paucity of replication origins, fork stalling at AT-rich DNA sequences, late replication timing, and collision of replication and transcription, which are all associated with perturbation of DNA replication[5,44]. Our study reveals new mechanisms involving concerted roles of FANCM and Rad52 to maintain stability of structure-prone AT-rich DNA sequences at CFSs, especially upon replication stress and oncogene expression.

We identified a novel function of FANCM in protecting AT-rich sequences at CFSs. Such FANCM function requires its translocase activity and its binding partners FAAP24 and MHF. We propose that AT-rich sequences at CFSs form DNA secondary structures when ssDNA is exposed during replication, and FANCM is recruited by FAAP24 and MHF to these DNA sequences. FANCM promotes fork reversal through its translocase activity to remove secondary structures formed at AT-rich sequences, thereby preventing DSB formation at CFSs (Fig. 7, left, pathway 1), which is supported by accumulation of γH2AX at Flex1 site and at endogenous FRA3B locus in FANCM-deficient

**Fig. 2** FAAP24 and MHF are important for recruiting FANCM to Flex1 to prevent DSB formation and mitotic recombination. **a** DSB formation at Flex1 or Luc surrounding regions was analyzed by ChIP analysis of pCEP4-Flex1 or pCEP4-Luc[10] using H2AX-S139p (γH2AX) antibody. Anti-γH2AX ChIP was performed in U2OS cells expressing FANCM shRNA (left), expressing Flag-FANCM-WT or K117R with endogenous FANCM silenced by shRNA (middle), or expressing FAAP24 shRNA with or without HU treatment (2 mM, 6 h) (right). Western blots for the expression of FANCM, Flag-FANCM WT and K117R, and FAAP24 are shown in Supplementary Fig. 5b. **b** Flag-FANCM or empty vector (Ctrl) was expressed in U2OS (HR-Flex) reporter cell line with Flex1 stably integrated into the genome. Anti-Flag ChIP was performed using indicated primer sets (P1 or P2), or primers corresponding to GAPDH (left); or using P1 before or after APH treatment (0.4 μM, 24 h) (right). ChIP value in Ctrl is set up as 1 for normalization. **c** Anti-Flag ChIP at Flex1 was performed in U2OS (HR-Flex) cells expressing Flag-FANCM and shRNAs for FAAP24 (left) or MHF1 (right) with vector as a control. Western blots for the expression of FAAP24 and MHF1 are shown in Supplementary Fig. 5d. **d** Spontaneous recombination was assayed in U2OS (HR-Flex) cells expressing shRNA for FAAP24 (left) or MHF1 (right) with a vector control. **e** Anti-Flag ChIP at Flex1 was performed in U2OS (HR-Flex) cells expressing Flag-FANCM-WT or indicated mutants with the vector as a control. Western blots showing FANCM expression are in Supplementary Fig. 5e. **f** Anti-Flag ChIP at Flex1 was performed in U2OS (HR-Flex) cells expressing Flag-FAAP24 WT or mutants (left), or Flag-FANCM and HA-FAAP24 WT or indicated FAAP24 mutants with endogenous FAAP24 silenced by shRNA (right). ChIP in the vector expressing cells serves as the control. DB DNA binding mutant: K171A/K173A; MB FANCM binding mutant: V198A/V199A/G200A. Western blots showing relevant protein expression are in Supplementary Fig. 5f, g. **g** Spontaneous recombination was assayed in U2OS (HR-Flex) cells expressing Flag-FAAP24 WT and mutant alleles with endogenous FAAP24 silenced by shRNA

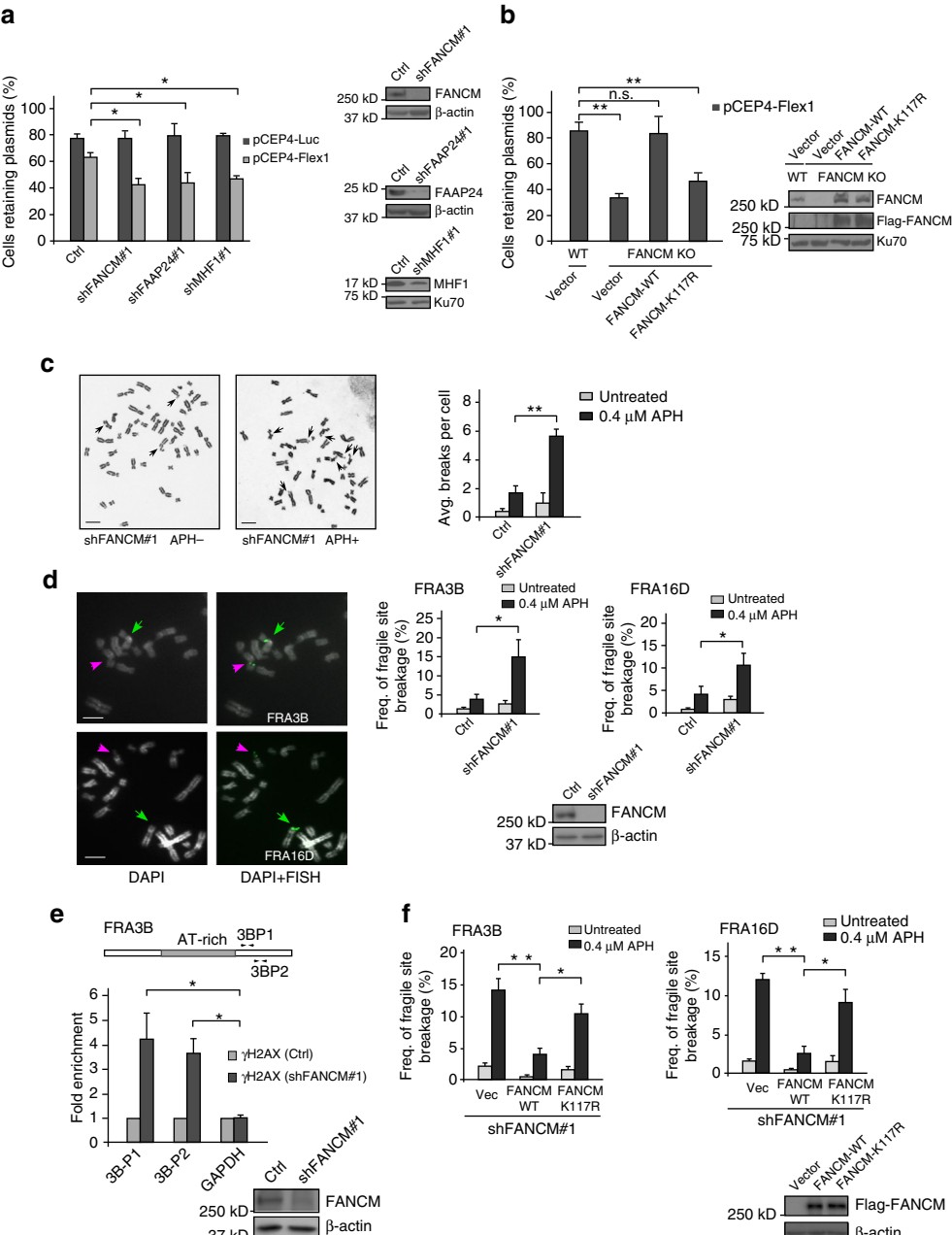

**Fig. 3** FANCM translocase activity is important for CFS protection. **a**, **b** Plasmid stability assay was performed in cells carrying pCEP4-Flex1 or pCEP4-Luc plasmids after culturing cells without hygromycin for 1 week. U2OS cells expressing indicated shRNAs or control vector (**a**), and HCT116 WT or FANCM KO cells reconstituted with Flag-FANCM (WT or K117R) (**b**) were used. **c** Metaphase spread of HCT116 cells was performed with or without expressing FANCM shRNA or vector before and after APH treatment (0.4 μM, 24 h). Representative images of metaphase spread are shown on the left, and overall chromosome gaps and breaks per cell are shown on the right. **d** FISH images of HCT116 cells after DAPI staining using probes against FRA3B or FRA16D are shown (left). Purple and green arrows indicate broken and normal chromosomes, respectively. Frequency of CFS expression at FRA3B or FRA16D in HCT116 cells expressing FANCM shRNA or control vector before and after APH treatment was determined (right). FANCM expression is shown by western blots. **e** Anti-γH2AX ChIP analysis at FRA3B locus using primer sets 3B-P1 and 3B-P2 was performed in U2OS cells expressing FANCM shRNA or vector before and after APH treatment (0.4 μM, 24 h). ChIP value in Ctrl is set up as 1 for normalization. **f** Frequency of CFS expression at FRA3B or FRA16D in HCT116 cells expressing Flag-FANCM (WT or K117R) with endogenous FANCM silenced before and after APH treatment (0.4 μM, 24 h) was determined. Expression of indicated Flag-FANCM alleles is shown by western blots. Bars, 5 μm

cells. Normal forks can be regenerated and restarted upon fork restoration.

FANCM binds to DNA with a preference for branched structures, such as Holliday junctions, replication forks, and D-loops[20,24]. FANCM binding partners FAAP24 and MHF stimulate FANCM-DNA binding activity[17,18,25]. We showed that the recruitment of FANCM to stalled replication forks at Flex1

requires both FAAP24 and MHF. Biochemically, FANCM exhibits ATP-dependent branch point translocase activity, which promotes replication fork regression[23,24], and such fork remodeling activity is stimulated by MHF but is independent of other FA proteins[17,25]. We showed that the translocase activity of FANCM but not the FA core complex is important for maintaining Flex1 stability, supporting the model that the key role of

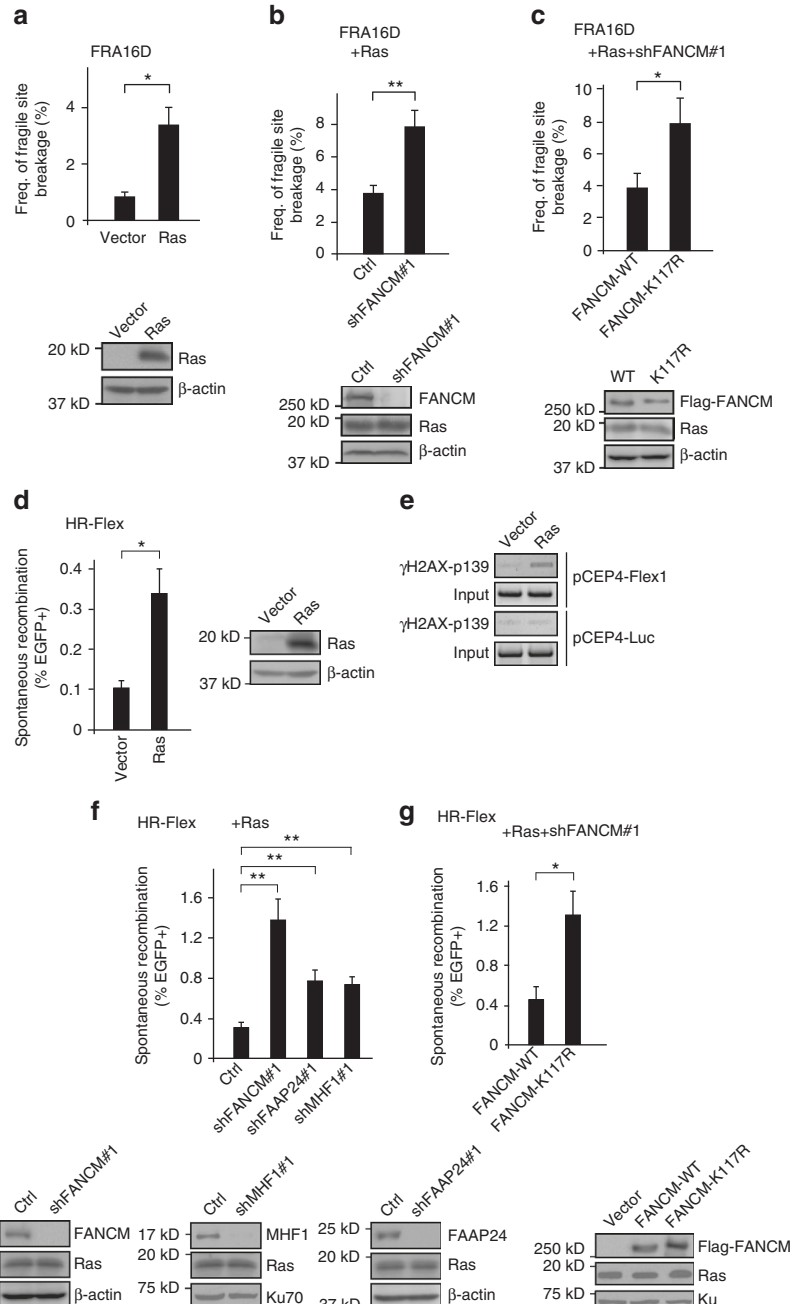

**Fig. 4** FANCM is important for suppressing Ras-induced CFS expression and mitotic recombination at Flex1. **a–c** Frequency of CFS expression at FRA16D in HCT116 cells was determined after expressing H-Ras V12 (Ras) or vector (**a**), Ras along with FANCM shRNA or control vector (**b**), or Ras and Flag-FANCM (WT or K117R) with endogenous FANCM silenced by shRNA (**c**). Expression of Ras and FANCM is shown by western blot analysis. **d, f, g** Spontaneous recombination was determined in U2OS (HR-Flex) cells with p53 silenced by shRNA (to prevent Ras-induced senescence) 1 week after Ras expression (**d**), Ras expression following expression of indicated shRNAs or vector (**f**), or Ras and Flag-FANCM WT or K117R with endogenous FANCM silenced by shRNA (**g**). Indicated protein expression is shown by western blot analysis. **e** ChIP analysis of γH2AX at Flex1 was performed in U2OS cells containing pCEP4-Flex1 or pCEP4-Luc plasmids 72 h after Ras overexpression

FANCM in preserving Flex1 stability is through its translocase activity, which removes DNA secondary structures at stalled replication forks.

ATR activation is critical for replication fork protection and CFS maintenance[45]. It has been shown that the FANCM/FAAP24 complex is associated with the checkpoint protein HCLK2 and facilitates ATR activation[46–48]. However, while acute inactivation of FAAP24 or FANCM by siRNAs both lead to an ATR checkpoint defect[46,48], only KO of FAAP24 but not FANCM in HCT116 impairs ATR signaling, possibly due to adaptation of

cells to the loss of FANCM[38]. In FANCM KO HCT116 cells, mitotic recombination at Flex1 is significantly increased compared to wild-type cells, revealing a defect in Flex1 protection in the presence of normal ATR signaling (Supplementary Fig. 3b). Thus, the role of FANCM in Flex1 protection is not primarily through facilitating ATR activation, but rather is mediated directly by its fork remodeling activity to remove DNA secondary structures.

Multiple components of the FA pathway have been linked to CFS protection. Besides FANCM, loss of FA proteins such as FANCA,

FANCB, and FANCD2 and its downstream BRCA1 also has been shown to increase CFS expression[14,49]. Our designed HR-Flex reporter specifically monitors Flex1-induced mitotic recombination by HR. If DSBs are formed at Flex1 in the reporter and are repaired by HR, the EGFP open reading frame would be restored to produce

green cells. We showed that different from loss of FANCM, inactivation of FANCA and FANCD2 does not significantly increase Flex1-induced mitotic recombination, while inactivation of BRCA1 reduces Flex1-induced mitotic recombination. This is consistent with the notion that different FA proteins have distinct roles to

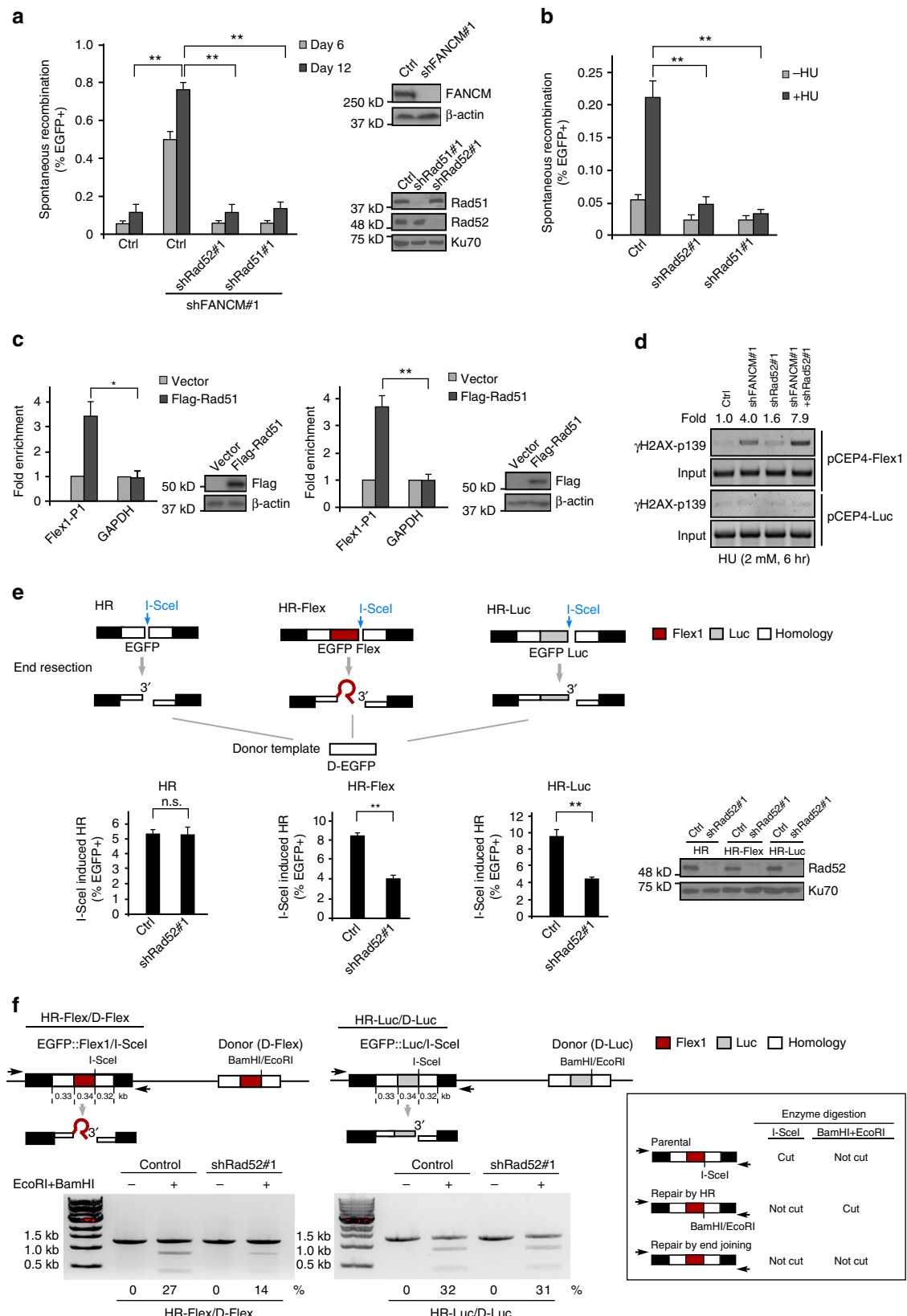

protect CFSs. BRCA1 is required for HR-mediated DSB repair at Flex1, and its checkpoint function is also needed for CFS protection[49]. FANCD2 and FANCI are specifically associated with CFS foci and are involved in preventing fragile site anaphase bridging[26]. FANCD2 was also found to facilitate replication through CFSs by suppression of DNA:RNA hybrid formation and by influence of dormant origin firing[50]. The function of FANCM to preserve stability of AT-rich sequences at CFSs through its fork remodeling activity is distinct from these previously described roles of FA proteins in CFS protection.

We also showed that the FANCM-MM1 mutant impaired in the interaction with the FA core complex is not defective in suppressing Flex1-induced mitotic recombination, which is consistent with the observation that the FA core complex is not required for maintaining Flex1 stability. However, the FANCM-MM1 mutant still exhibits increased CFS expression. It was shown that the interaction of FANCM and the FA core complex is important for ID2 recruitment and mono-ubiquitination[34], and thus the defect of FANCM-MM1 mutant in CFS protection is likely related to modulating the function of the FA core complex and ID2, which is in line with the previous findings that the FA core proteins and ID2 are important for CFS protection[14]. Collectively, the FA network employs multiple mechanisms to protect CFSs, and our study reveals a novel function of FANCM to remodel forks at the sites containing DNA secondary structures through its translocase activity to maintain CFS stability.

When FANCM is deficient, HR-mediated mitotic recombination is significantly increased to repair DSBs accumulated at Flex1. This suggests that HR is used as a backup mechanism to repair DSBs accumulated at structure-prone DNA sequences at CFSs when FANCM function is impaired (Fig. 7, left, pathway 2). Interestingly, we found that not only Rad51 but also Rad52 is important for suppression of hyper-mitotic recombination induced upon loss of FANCM. In yeast, Rad52 plays a prominent role in all types of HR[30,51], but in mammalian cells, Rad52 is not essential for general HR such as gene conversion[29]. Powell's group showed that when BRCA2 or several other HR players are inactivated, Rad52 is needed for HR and cell growth[39,52,53]. Human Rad52 exhibits DNA annealing activity in vitro, but its mediator activity to promote Rad51 filament formation and strand invasion is largely replaced by BRCA2[54]. It was proposed that Rad52 may delineate an alternative backup mediator pathway and function in the absence of BRCA2[29].

In the HR reporter (HR), both DSB ends share homology to the donor sequence (except 12 bp at the left side and 13 bp at the right side due to the insertion of the I-SceI site), and gene conversion frequency is not obviously altered when Rad52 is depleted (Fig. 5e). However, in the HR-Flex or HR-Luc reporter where a 0.34 kb Flex1 or a similar-sized luciferase sequence non-homologous to the template is present at the DSB, HR is significantly reduced in Rad52-deficient cells. We further showed that even with perfect homology placed in the donor templates, Rad52 is still required when CFS-derived AT-rich sequence Flex1 is present at the DSB ends (Fig. 5f), which is predicted to form DNA secondary structures after end resection. We thus propose that Rad52 is required for HR when DSB ends are blocked by nonhomologous sequences or DNA secondary structures.

Multiple mechanisms can be proposed for this special requirement of Rad52. First, Rad52 may be required to initiate and/or facilitate Rad51-mediated strand invasion from a blocked 3′ end (Supplementary Fig. 11, left). We speculate that Rad52 helps Rad51 initiate strand invasion from a blocked end by using its ssDNA annealing function to anneal the 3′ blocked strand to the template when the dsDNA of the template is transiently opened. Alternatively, Rad52 may use its newly identified inverse strand exchange activity[55] to initiate the pairing of the invading strand from the blocked end with the template strand. As another possible mechanism, Rad52 may be required for second-end capture of the blocked end if strand invasion occurs from the other homologous end (Supplementary Fig. 11, right). Rad52 ssDNA annealing activity is believed to mediate second-end capture during synthesis-dependent strand annealing (SDSA) in yeast[30], but such activity is not required in mammalian cells, since HR is not significantly altered in Rad52-deficient cells. However, when the end is blocked, the annealing activity of Rad52 may become indispensable for second-end capture. The possible roles of Rad52 in promoting strand invasion and in capturing the second-end from a blocked DSB end may not be mutually exclusive, and Rad52 could be involved in both. After Rad52-mediated pairing of the 3′ blocked strand with the template, unpaired DNA tails will be removed by endonuclease cleavage in a manner similar to single strand annealing (SSA) mechanism[56], and DNA synthesis will start to complete HR (Supplementary Fig. 11).

When sister chromatids are used as templates for HR-mediated DSB breaks, no non-homologous sequences are present in the sister chromatid templates. However, when replication forks are collapsed at Flex1 or at other structure-prone DNA sequences, single-ended DSBs would be blocked by DNA secondary structures formed after end resection, and under such condition, Rad52 is required for strand invasion to promote replication restart (Fig. 7, middle). If a converging fork arises from the other side, two-ended DSBs with one blocked end would require Rad52 either for strand invasion or for second-end capture (Fig. 7, right). Therefore, while Rad52 is nonessential for general HR, it becomes indispensable for HR when DSBs are generated at special chromosomal regions such as CFSs.

**Fig. 5** Rad52 plays a critical role in repairing DSBs at Flex1. **a**, **b** Spontaneous recombination was determined in U2OS (HR-Flex) cells expressing shRNAs for Rad51 or Rad52, or control vector, 6 or 12 days after expression of FANCM shRNA or control vector (**a**), or 4 days after HU treatment (2 mM, 24 h) or without (**b**). **c** Anti-Flag ChIP at Flex1 was performed in U2OS (HR-Flex) cells expressing Flag-Rad51 (left) and Flag-Rad52 (right). The expression of indicated protein is shown by western blots. **d** ChIP analysis of γH2AX at Flex1 or Luc was performed in U2OS cells containing pCEP4-Flex1 or pCEP4-Luc plasmids and expressing shRNAs for FANCM, Rad52 or both after treatment of HU (2 mM, 6 h). The ChIP signal of γH2AX is quantified by relative fold to the Ctrl. **e** Schematic drawing of HR reporters containing the I-SceI site (HR), Flex1 with the I-SceI site (HR-Flex), and Luc with the I-SceI site (HR-Luc) is shown (top). The homology of the I-SceI cleaved EGFP cassettes to the donor template (D-EGFP) is indicated as white boxes. I-SceI-induced HR was assayed in U2OS HR, HR-Flex, and HR-Luc reporter cell lines with expression of Rad52 shRNA or control vector (bottom). Rad52 expression is shown by western blot analysis. **f** Schematic drawing of HR reporters HR-Flex/D-Flex and HR-Luc/D-Luc, which contain identical Flex1 and Luc sequences in both EGFP recipient cassettes (EGFP::Flex1/I-SceI and EGFP::Luc/I-SceI) and the donor templates (D-Flex or D-Luc) is shown on top. The BamHI and EcoRI sites in the donor templates are at the corresponding position of the I-SceI site in the recipient EGFP cassettes. U2OS (HR-Flex/D-Flex) or U2OS (HR-Luc/D-Luc) cells with or without expressing Rad52 shRNAs were infected with I-SceI lentiviruses and 3 days after, genomic DNA was extracted and digested with I-SceI, followed by PCR using indicated primers (shown as arrows). The percentage of BamHI and EcoRI digestible PCR products among total DNA was calculated. Predicted enzyme digestion patterns of parental EGFP recipient cassettes (EGFP::Flex1/I-SceI and EGFP::Luc/I-SceI) and the repair products generated by HR or imperfect end joining are shown in the box at right

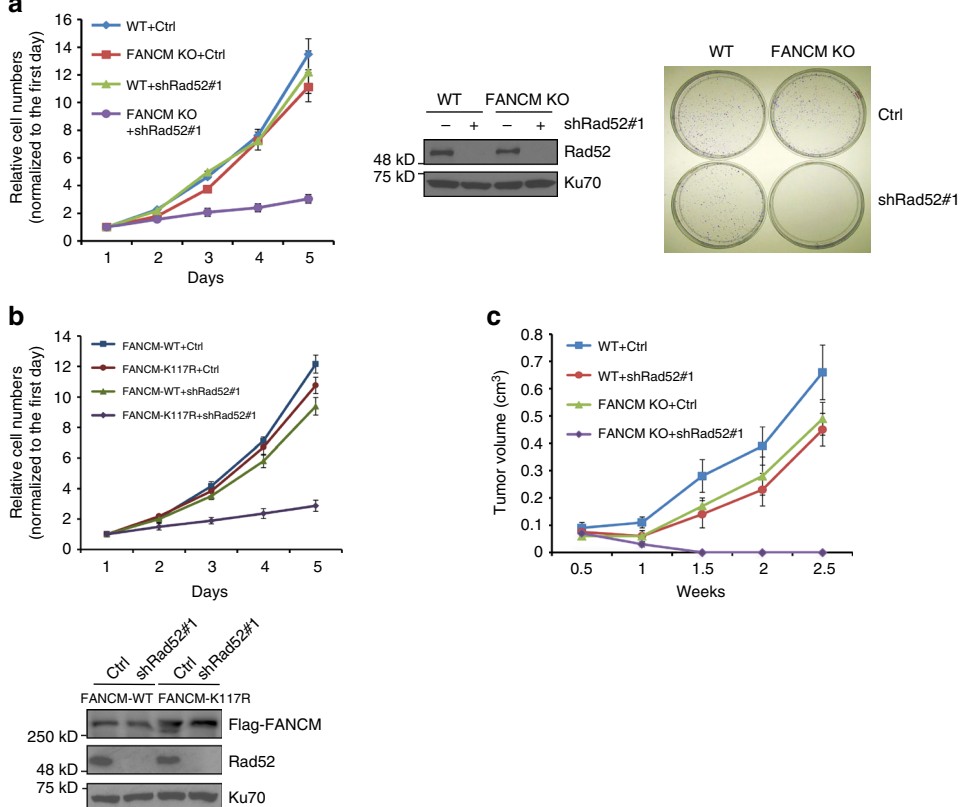

**Fig. 6** Inactivation of Rad52 drastically reduces the growth of FANCM KO cells and tumors. **a**, **b** Growth curve of HCT116 WT or FANCM KO cells was plotted after expressing Rad52 shRNA or control vector (**a**, left), or FANCM KO cells expressing FANCM WT or K117R followed by expression of Rad52 shRNA or control vector (**b**). The viability of HCT116 WT or FANCM KO cells expressing Rad52 shRNA or vector was analyzed by colony formation assay (**a**, right). **c** Tumor volume was measured in nude mice after tumor inoculation with HCT116 WT or FANCM KO cells expressing Rad52 shRNA or vector. Data are shown as the mean ± SD of tumor volumes of five mice in each group at each time point

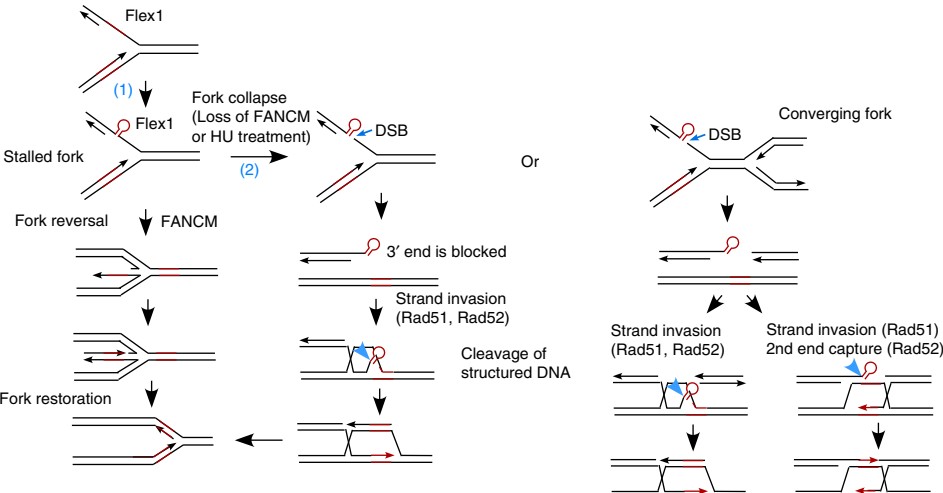

**Fig. 7** Proposed model for concerted roles of FANCM and Rad52 in protecting Flex1. Blue-marked (1) and (2) refer to pathway 1 (FANCM-mediated fork reversal to remove DNA secondary structures) and pathway 2 (Rad51/Rad52-dependent HR to repair DSBs accumulated at structure-prone DNA sequences when FANCM is deficient), respectively. See Discussion for more details

Rad52 was shown to be retained at CFSs on mitotic chromatin and is important for mitotic DNA synthesis by recruiting Mus81 and PolD3 to CFSs in early mitosis[57]. Rad52 was also found to be important for promoting BIR upon replication stress[58]. In the context of CFSs, these functions of Rad52 are also expected to be important and may be used cooperatively with the new role that

we identified—mediating HR at structure-prone DNA sequences to preserve CFS stability.

Our study reveals the important and concerted roles of FANCM and Rad52 in repairing DSBs at CFSs and preserving CFS stability. In a normal cell, FANCM protects structure-prone DNA sequences by promoting fork reversal using its translocase activity (Fig. 7,

pathway 1), thereby preventing DSB formation at CFSs, and thus Rad52 is not essential. However, when FANCM is deficient, DSBs are accumulated at structure-prone DNA sequences, which require Rad51/Rad52-dependent HR for repair, and thus Rad52 becomes indispensable (Fig. 7, pathway 2 and Supplementary Fig. 11). This is supported by the observation that inhibition of Rad52 expression in FANCM KO cells results in a significant reduction in cell growth, although neither *FANCM* nor *Rad52* are essential genes[33,40]. These studies suggest a synthetic lethality interaction between FANCM and Rad52, and Rad52-dependent HR acts as a critical backup pathway for FANCM-mediated genome protection function. We also speculate that FANCM and Rad52 are important not only for protecting CFSs, but also for maintaining the stability of various kinds of DNA sequences that are prone to secondary structure formation throughout the genome. The synthetic lethality interactions between these two genes may reflect their more general roles beyond CFS protection.

Rad52 is synthetically lethal with BRCA2 deficiency[39]. However, in contract to an essential function of BRCA1 and BRCA2 in HR[59], FANCM is only minimally involved in HR. In this respect, disruption of *BRCA1* or *BRCA2* in mice leads to early embryonic lethality[60], but *FANCM* KO mice develop normally[40]. In addition, loss of FANCM leads to a substantial increase of HR-mediated mitotic recombination at Flex1, whereas deficiency of either BRCA1 or BRCA2 suppresses Flex1-induced HR (Fig. 1b and Supplementary Fig. 2a and 2b). Thus, the synthetic lethality interaction between FANCM and Rad52 is distinct from the genetic interaction of Rad52 with BRCA1/BRCA2.

FANCM has been identified as a breast cancer susceptibility gene, and its deficiency is associated with breast cancer, especially triple-negative breast cancer[43,61,62]. Mutations in FANCM confer a predisposition to high grade serous ovarian cancer[41]. Down-regulation of FANCM was also described in sporadic head and neck squamous cell carcinoma[42]. Our study identifies an essential function of Rad52 in FANCM-deficient cells. Therefore, inhibition of Rad52 can serve as a new targeted therapeutic strategy for treating FANCM-deficient tumors, which is expected to have low toxicity to normal cells.

## Methods

**Cell cultures**. U2OS, T98G, MCF7, Hela, HCT116, and 293T cells were from ATCC and cultured at 37 °C and 5% $CO_2$ in Dulbecco's modified Eagle's medium with 10% fetal bovine serum in the presence of antibiotics. Cell lines tested negative for mycoplasma contamination. All reporter cell lines were verified by Southern blot analysis. FANCM knockout HCT116 cells were previously described[38].

**Plasmid construction and recombinant protein expression**. FANCM wild-type and indicated mutants were generated by PCR and subcloned into NBLV0051 (Novo Bio) vector containing a 3× N-terminal Flag-tag. Small hairpin RNA (shRNA) target site-resistant mutations were generated by site-directed mutagenesis (QuikChange, Agilent). FAAP24, MHF1, Rad51, and Rad52 were subcloned into pBabe-Puro, with 3× N-terminal Flag-tag. K171A/K173A and V198A/V199A/G200A mutations of FAAP24, and shRNA-resistant mutations of FAAP24 and MHF1 were generated by site-directed mutagenesis. Stable expression of exogenous proteins was generated by lentiviral or retroviral infection followed by appropriate drug selection.

pCEP4-Flex1 and pCEP4-Luc plasmids were constructed previously[10]. The EGFP-based HR reporter constructs including the HR reporter containing the I-SceI cleavage site, the HR-Flex reporter containing the 0.34 kb Flex1 and the I-SceI cleavage site, and the HR-Luc reporter containing a 0.34 kb luciferase-derived fragment and the I-SceI cleavage site, were constructed previously[10,63].

DNA fragments containing AT-rich sequences 16C/AT1 (486 bp, genomic position chr16: 65356038–65356522, GRCh37.p13 Primary Assembly) and 16C/AT3 (448 bp, genomic position chr16: 65570672–65571120) from FRA16C, and AT-rich sequence 3B/AT (440 bp, genomic position chr3: 60846416–60846856) from FRA3B were synthesized by GeneArt Gene Synthesis (Thermo Fisher Scientific) and subcloned into the HR reporter to generate HR-16C/AT1, HR-16C/AT3 and HR-3B/AT reporters, in a similar way as the HR-Flex reporter was constructed[10].

To construct the HR-Flex/D-Flex and HR-Luc/D-Luc reporters, the donor templates in the original HR-Flex and HR-Luc reporters[10] were replaced with

N- and C-terminal truncated EGFP cassettes carrying Flex1 and Luc sequences respectively, which are identical to the I-SceI-containing recipient cassettes (EGFP::Flex1/I-SceI and EGFP::Luc/I-SceI, Fig. 5f) except that the corresponding I-SceI site is replaced with the BamHI/EcoRI sites.

**shRNA interference**. Silencing of indicated endogenous genes was performed by retroviral or lentiviral infection using pMKO and pLKO vectors, respectively, to express corresponding shRNAs[64,65]. shRNA target sequences were designed by Dharmacon and listed below. FANCM, #1 GAACAAGAUUCCUCAUUACUU and #2 CAAACCAUGUUCACAAUUAGA; FANCD2, #1 GGUCAGAGCUGU AUUAUUC and #2 GAUAAGUUGUCGUCUAUUA; FANCA, #1 CGACAUGCA UGCUGUGGGAUC, #2 CGCUUUUGGCUGCUGGAGUACA; FAAP24, #1 CCGGAUGAGUGAACAAUACUU, #2 AUUUUCGAGGAUGGCUUGACA; MHF1, #1 CCAUUUCGGAGCUGACUUUCC, #2 CGAAAAGAACCACAA UUAACA; Rad51, #1 CUAAUCAGGUGGUAGCUCAAG, #2 GAAGCUAUGU UCGCCAUUAAU; Rad52, #1 GGAUGGUUCAUAUCAUGAAGA, #2 GAU GUUGGUUAUGGUGUUUAGU; BRCA1, #1 CAACAUGCCCACAGAUCAACU and #2 CCAAAGCGAGCAAGAGAAUCU; Mre11, #1 GAUGAGAACUCU UGGUUUUAAC and #2 GAGUAUAGAUUUAGCAGAACA; CtIP, #1 GAGCAGA CCUUUCUCAGUAUA and #2 GCUAAAACAGGAACGAAUCUU. BRCA2 shRNA sequence GAAGAAUGCAGGUUUAAUA[66]. Uncropped blots are shown in Supplementary Fig. 12.

**Mitotic recombination assay and HR assay after I-SceI induction**. For spontaneous mitotic recombination assay, cells were pre-sorted by fluorescence-activated cell sorting (FACS) to remove previously accumulated green cells and cultured for indicated days[10]. The mitotic recombination frequency was determined by FACS analysis. For I-SceI-induced HR assay, reporter cell lines were infected with retroviruses encoding HA-I-SceI, and EGFP-positive events were scored by FACS analysis 5 days later. FACS analysis was performed using a BD Accuri C6 flow cytometer and accompanying data analysis software (CFlow, Becton-Dickinson).

To analyze HR usage in U2OS (HR-Flex/D-Flex) and U2OS (HR-Luc/D-Luc) cells, genomic DNA was extracted 3 days after I-SceI viral infection and digested with I-SceI, followed by PCR using primers indicated in Fig. 5f (GGATAGCGGTTTGACTCA CGGGG and TTACTTGTACAGCTCGTCCATGC). PCR products were digested with or without BamHI and EcoRI and resolved by electrophoresis. The percentage of BamHI and EcoRI digestible PCR products among total DNA was calculated after quantifying DNA bands using Image J.

**Plasmid stability assay**. pCEP4-Flex1 or pCEP4-Luc plasmids contain the Epstein-Barr virus (EBV) replication origins and propagate as episomes in mammalian cells[10,67]. Plasmid stability of pCEP4-Flex1 or pCEP4-Luc in U2OS or HCT116 cells was determined as described[10]. Briefly, the plasmids were transfected into U2OS or HCT116 cells by lipofectamine 2000 (Thermo Fisher Scientific), followed by hygromycin selection, and then the plasmid-containing cells were cultured in the absence of hygromycin for 1 week. The percentage of cells retaining the plasmids after 1 week without hygromycin selection was determined by the number of hygromycin-resistant cells over total number of cells.

**Growth curve and cell viability assay**. Cell proliferation in HCT116 and its derivative cell lines was measured by hemocytometer counting of trypsinized cells every 24 h. Cell number was normalized to day 1. Colony formation of HCT116 cells and its derivative cell lines was used to show cell viability. Cells were plated at 5000 cells per 10 cm plate, and were grown in complete media for 8 days, followed by fixing with cold methanol and staining with 1% crystal violet.

**Immunoblotting and antibodies used**. Whole cell lysis and immunoblotting were performed as described[63,68]. Antibodies against FANCM and FAAP24, and antibody against BRCA2 were kindly provided by Dr. Weidong Wang[38] and Dr. Jun Huang[66], respectively. Antibodies against Mre11 and CtIP were used previously[10,63,69]. Commercial antibodies used were: Anti-FANCA (Bethyl Laboratories A301-980A,1:1000), anti-FANCD2 (Bethyl Laboratories A302-174A, 1:1000), anti-MHF1 (Abcam, ab169385, 1:1000), anti-Rad51 (Santa Cruz Biotechnology, sc-398587, 1:500), anti-H2AX-S139p (Cell Signaling, #2577, 1:1000), anti-H2A (Cell signaling, #2578, 1:1000), anti-FLAG (Sigma, F1804, 1:2000), anti-Rad52 (Santa Cruz Biotechnology, sc-365341, 1:1000), anti-Ras (Santa Cruz Biotechnology, sc-520, 1:1000), anti-β-actin (Sigma, A5441, 1:2000), anti-Chk1 (Santa Cruz Biotechnology, sc-8408, 1:500), anti-Chk1-S345p (Cell Signaling, #2348, 1:500), anti-RPA32 (Bethyl Laboratories A300-244A, 1:1000), BRCA1 (Bethyl Laboratories, A300–000A, 1:500) and FITC-conjugated anti-mouse IgA (BD, 559354).

**Chromatin immunoprecipitation**. Chromatin immunoprecipitation (ChIP) was performed as described with some modifications[10,67]. Cells were fixed with 1% formaldehyde for 10 min at room temperature and the reaction was stopped with 1.25 M glycine solution. After washing twice with cold PBS, cells were resuspended in lysis buffer (1% SDS, 10 mM EDTA, 50 mM Tris–HCl, pH 8.1) supplemented

with protease inhibitor cocktail ("PIC", cOmplete, Roche) and subject to sonication to break chromatin into fragments with an average length of 0.2–1.0 kb. The lysate supernatant was pre-cleared with Protein A/G Sepharose beads (Amersham Biosciences). IP was performed using H2AX-S139p (Cell Signaling #2577) or Anti-FLAG antibody (Sigma F1804) followed by washing with 1 ml TSE I (0.1% SDS, 1% Triton X-100, 2 mM EDTA, 20 mM Tris–HCl, pH 8.1, 150 mM NaCl) with PIC, TSE II (0.1% SDS, 1% Triton X-100, 2 mM EDTA, 20 mM Tris–HCl, pH 8.1, 500 mM NaCl) with PIC, buffer III (0.25 M LiCl, 1% NP-40, 1% deoxycholate, 1 mM EDTA, 10 mM Tris–HCl, pH 8.1) with PIC, and TE with PIC. The protein–DNA complex was eluted from beads by elution buffer (1% SDS, 0.1 M NaHCO$_3$), and cross-linking was reversed by adding in 4 µl of 5 M NaCl and incubating at 65 °C for 6 h, followed by proteinase K digestion for 2 h at 42 °C. DNA was purified by QIAquick kit (QIAGEN) according to the manufacturer's instructions. For ChIP at Flex1 in the HR reporter stably integrated in the genome or at AT-rich sequences in the endogenous FRA3B locus, recovered DNA was analyzed by RT-PCR and the readout was normalized to the vector control which is set as 1. GAPDH locus was used as a control to show the specificity of protein binding to Flex1. The primers used for ChIP at Flex1 in HR reporter: P1F 5′ CTCCAATTCGCCCTATAGTGAGTCGTATTA, P1R 5′TTACTTGTACAGCTC GTCCATGC, P2F 5′GGCAGTACATCAATGGGCGTG, and P2R 5′CCTTTA GTGAGGGTTAATTGCGCG; at AT-rich sequences in FRA3B: 3B-P1F 5′TT AGCCTACTTCAGGGTTTCT, 3B-P1R 5′TGGAGAGGTTACTACTGGCA, 3B-P2F 5′TATGGAGGGCTGTCCTATGC, 3B-P2R 5′TGATGATAGCAATGATGG TGATG; and at GAPDH: GAPDH-F: 5′CCCTCTGGTGGTGGCCCCTT, GAPD H-R: 5′GGCGCCCAGACA CCCAATCC. For ChIP at Flex1 or Luc surrounding regions in the pCEP4-Flex1 or pCEP4-Luc plasmids, the recovered DNA was amplified by regular PCR with primers 5′TCAGGGGGAGGTGTGGGAGG and 5′ GCAGTCCACAGACTGCAAAG, and PCR products were resolved by 1.5% agarose DNA gel[10].

**Metaphase chromosome analysis and fluorescence in situ hybridization.** Metaphase chromosome analysis and FISH were performed as described[10]. Cells were treated with 0.4 µM APH for 18 h, followed by treatment of 0.1 µg/ml colcemid at 37 °C for 75 min. Cells were then collected and resuspended in 75 mM KCl hypotonic solution pre-warmed to 37 °C and incubated at 37 °C for 30 min, followed by several changes of fixative solution (3:1 methanol/acetic acid). Cells were dropped onto slides and incubated for 2 h at 60 °C prior to Giemsa staining or FISH analysis. Breaks and gaps were quantified on Giemsa-stained metaphases. Fifty metaphases per sample were scored for the number of overall chromosome gaps and breaks. FISH experiments were performed according to standard protocols[70]. Green 5-Fluorescein dUTP-labeled probes 264L1 (FRA16D) and 641C17 (FRA3B) from RPCI-11 human BAC library (Empire Genomics) were used as probes for FISH analyses. Chromosomes were counterstained with DAPI.

**Mouse models.** In the mouse tumor xenograft analysis, mice were randomly allocated into experimental groups, and the experimenter was blinded to the group allocation. HCT116 cells or FANCM-knockout HCT116 cells expressing Rad52 shRNA or the control vector were injected subcutaneously into 6-week-old female BALB/c-nu mice ($n = 5$ per group). Each mouse received subcutaneous injection of $10^6$ cells in 100 µl PBS. Tumor volume was measured twice per week using calipers, and tumor volume was calculated as length × (width$^2$) × 0.5. Data are represented as the mean ± SD of five mice for each experiment.

All animal experiments were conducted at Anticancer Biotechnology (Beijing) Co. Ltd. Tumor xenograft experiments were performed with the approval of the China Committee for Research and Animal Ethics in compliance with the guidance on experimental animals.

**Quantification and statistical analysis.** Excel was used for the statistical analyses. Two-tailed nonpaired parameters were applied in TTEST (Student's $t$ test) to analyze the significance of the differences between samples.

**Data availability.** All data generated and analyzed during this study are included in this published article and its Supplementary Information files.

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

## Acknowledgements

FANCM wild-type and mutant-containing plasmids were kindly provided by Dr. Stephen C. West (The Francis Crick Institute, UK) and Dr. Weidong Wang (NIH). Plasmids containing wild-type FAAP24, MHF1, Rad51, or Rad52 were provided by Dr. Stephen C. West, Dr. Yong Xiong (Yale University), Dr. Jeremy Stark (City of Hope), and Dr. Patrick Sung (Yale University), respectively. We thank Dr. Weidong Wang and Dr. Jun Huang (Zhejiang University) for providing anti-FANCM and anti-FAAP24 antibodies and BRCA2 antibody, respectively and Dr. Catherine Freudenreich (Tufts University) for sharing the Flex1-containing plasmid. The shRNA vectors pMLO.1-puro (#8452), pLKO.1-TRC (#10878), and pLKO.1-blast (#26655) are from Addgene. This work is supported by NIH grants CA187052, CA197995, and GM080677 to X.W.; National Basic Research Program of China (2015CB910602) and National Natural Science Foundation of China (31370841) to H.W.; NIH grants CA179441 and CA193124-Project 3 to L.L.

## Author contributions

H.W., S.L., and X.W. designed and performed experiments, and analyzed the data; J.O. performed experiments and assisted in manuscript preparation; J.R. performed experiments and analyzed data; L.L. provided FANCM KO cell lines and gave valuable input to the manuscript; X.W. is responsible for the project's planning and experimental designing. X.W., H.W., and S.L. wrote the manuscript.

## Additional information

**Competing interests:** The authors declare no competing interests.

