## [Peer Review File · Nature Communications]

Reviewers' Comments:

Reviewer #1:

Remarks to the Author:

Fanconi anemia proteins have been previously implicated in the protection of common fragile sites (CFSs). In this manuscript, the authors used a Flex1/FRA16D derived reporter assay to characterize the function of FANCM in CFS protection. They found that FANCM, FAAP24 and MHF1/2, but not FANCD2/I, are important for suppressing recombination at the Flex1 reporter. The translocase activity of FANCM is also required. In FANCM deficient cells, the recombination at the Flex1 reporter is suppressed by Rad52 depletion, and the proliferation of FANCM deficient cells is inhibited by Rad52 knockdown both in vitro and in mouse. Overall the results in this manuscript provide many details of how FANCM functions at the Flex1 reporter. However, where these results can be generalized to various types of endogenous CFSs still needs further investigations.

Specific comments:

1. FANCD2 has been previously implicated in the protection of CFSs including FRA16D (Howlett et al. 2005 Human Mol Genet). Moreover, it has shown that FANCD2 is important for replication through CFSs including FRA16D (Madireddy et al. 2016 Mol Cell). BRCA1 is also well known to protect CFSs (Arlt et al. 2004 Mol Cell). It is very surprising that only FANCM but not other FA/BRCA proteins are required for suppression of recombination at the Flex1 reporter. The Flex1 reporter may not faithfully recapitulate the important features of CFSs.
2. In contrast to FRA16D, the fragility of FRA3B is thought to arise from low origin firing instead of DNA secondary structures and fork stalling (Letessier et al. 2011 Nature). It is very surprising that FANCM knockdown also increased the expression of FRA3B in Fig. 3.
3. In Fig. 4, how Ras induces recombination at FRA16D and the Flex1 reporter is not explained at all. Oncogene activation could induce replication stress in many different ways, and most of these are very indirect. Without an understanding of how Ras induces CFS expression, it is difficult to appreciate the function of FANCM in this context.
4. The results in Fig. 5a and 5b can be interpreted differently. Rad52 may be required for forming secondary structures, fork stalling, or DSB formation at the Flex1 reporter.
5. Based on the results in Fig. 5d, the authors proposed that Rad52 is important for HR when DSB ends are blocked by secondary structures. However, in the case of HR-Flex, DSB ends are blocked by both secondary structures and a non-homologous sequence. In the case of HR-Luc, DSB ends are blocked by a non-homologous sequence only. In neither case, DSB ends are blocked by secondary structures only. Therefore, the authors' hypothesis is not supported by experimental evidence.
6. The interpretation of the results in Fig. 6 may not be correct. FANCM is important for the response to replication stress in various situations, and its function is clearly not limited to CFSs. Even if FANCM deficient cells are dependent on Rad52 for proliferation, it does not necessarily mean that the proliferation defect of the double mutant arises from problems at CFSs. The dependency of FANCM deficient cells on Rad52 is virtually the same as the previously reported Rad52 dependency of BRCA2 defective cells (Zeng et al. 2010 PNAS). This dependency could be explained by the redundant HR functions of FA/BRCA proteins and Rad52.

Reviewer #2:

Remarks to the Author:

Previous studies have shown that several Fanconi anemia proteins, including components of the core complex and ID complex, accumulate at common fragile sites (CFS) and protect their

stability. In the current study, Wu and colleagues showed that FANCM, a key component of the FA pathway, also maintains CFS stability, but its impact is stronger than the core or ID complex. The authors showed convincingly that FANCM, together with its two DNA binding partners, is recruited to CFS and suppresses DSB formation and mitotic recombination. In addition, the translocase activity of FANCM is required for this function, but its interaction with FA core complex is dispensable. Interestingly, the authors found that RAD52 also stabilizes CFS, similar to FANCM. Depletion of RAD52 in FANCM-ko cells results in synthetic lethality. The authors conclude that FANCM and RAD52 act concertedly in protection of CFS. Targeting RAD52 may be a useful strategy to treat FANCM-deficient tumor.

This work is of high technical quality. The findings that FANCM and RAD52 act in concert to stabilize CFS should be of general interest to the DNA repair and cancer fields. I recommend its publication in Nature Communication after minor revisions described below.

1. The part of Introduction describing the conclusion is too long and overlaps with Discussion. It should be shortened.
2. Figure 1c and 1d are out of the order in citation in the text. This should be corrected.
3. Page 7, 2nd paragraph, the authors should make clear to readers that the DNA binding activity of FAAP24 is required for recruitment of FAAP24 to CFS, whereas the FANCM interacting motif is dispensable for the same recruitment. As written, it is unclear to reader whether the authors are describing recruitment of FAAP24 or FANCM.
4. RAD52 and FANCM are synthetic lethal, suggesting that the two proteins work in two parallel pathways to promote cell proliferation. This synthetic lethal interaction is different from their genetic interactions in CFS protection. The authors should make this clear to readers in the discussion.
5. Page 10, first paragraph, the authors stated that "these studies reveal an important function of RAD52 in mammalian cells in a special context when DNA ends are blocked by DNA secondary structure". I did not see evidence for this statement. Figure 5d shows that the luciferase construct has the same effect as the Flex report. Do the authors think that the luciferase reporter can also form secondary DNA structures? If this statement is just speculation, please move this part to the Discussion.

Reviewer #3:

Remarks to the Author:

This is an interesting paper describing the role of FANCM in common fragile site protection and FANCM-Rad52 synthetic lethality.

However, there are several concerns which the authors should address.

Major points:

- 1) Just one shRNA for each gene (FANCA, FANCD2, FAAP24, MHF1, RAD51, RAD52) was used for Figure 1b, 1c, Supplemental Figure 1c, Figure 5a,b,d. The authors should perform an "add-back experiment" by knocking down with shRNA followed by transfecting an expression plasmid to rescue the phenotype or use at least two shRNAs/gene. (For FANCM, the authors have performed the add-back experiments.)
- 2) Usage of just one cancer cell line (U2OS or HCT116) for most of the experiments is a concern, since whether the conclusion can be generalized is questionable. The authors should use multiple cell lines for some key experiments.
- 3) Figure 3e : the authors should test the effect of FANCM-MM1 mutant or depletion of other FA proteins (FANCA, FANCD2, etc.) on fragile site breakage.
- 4) Some statistical tests should be done for any quantitative data analyses.

Minor points:

- 1) Page 3 Line 21: "the FANCD2 and FANCI heterodimeric complex (ID2), which further recruits

the downstream FA proteins including FANCD1/BRCA2, FANCJ/BRIP1, FANCN/PALB2, and FANCO/RAD51 to DNA damage sites to promote HR” This sentence is incorrect. ID2 complex does not recruit BRCA2, BRIP1, PALB2 or RAD51C.

2) In reference 14 (Howlett, N.G., et al. Hum Mol Genet 14, 693-701 (2005)), it is reported that FANCD2 depletion in HCT116 leads to elevated fragile site instability following exposure to APH. The authors should comment on the interpretation of the discrepancy between ref 14 and the current paper.

Reviewer #1:

“Overall the results in this manuscript provide many details of how FANCM functions at the Flex1 reporter. However, where these results can be generalized to various types of endogenous CFSs still needs further investigations.”

Response: We acknowledge that multiple mechanisms underlie CFS instability, but fork stalling at unusual DNA sequences such as structure-forming AT-rich sequences is believed to be one of the general causes to induce CFS instability in addition to paucity of replication origin, large-sized open reading frames and late replication timing. The following points support the idea that FANCM-dependent protection of AT-rich structure-forming DNA sequences at CFSs can be regarded as a general mechanism for the maintenance of CFS stability.

1. Many CFSs contain structure-prone AT rich sequences which can stall DNA replication and cause DSB formation. In addition to Flex1 derived from FRA16D^{1,2}, AT-rich structure-forming DNA sequences are also found in FRA16C, FRA6E, FRA7H, FRA7G, FRA11G, FRA3, FRA2G and more³⁻⁶. Replication stalling has been observed at these sequences^{1-3,7}.
2. In this revision, we cloned another two AT-rich sequences from FRA16C and one from FRA3B, and showed that these sequences also induce mitotic recombination in a manner similar as Flex1 (new data in Supplementary Fig.1), suggesting that these sequences also cause genome instability as Flex1. Thus, induction of genome instability by CFS-derived AT-rich structure-forming DNA sequences is not limited to a specific AT-rich sequence Flex1, but likely a general mechanism applicable to other CFSs.
3. In normal cells, these AT-rich structure-forming DNA sequences are protected by the mechanisms such as FANCM-mediated fork reversal as we described, which minimizes the effect of DNA secondary structure-induced genome instability at CFSs. However, in the absence of such protection mechanisms, for example in FANCM deficient cells, more DSBs are generated at these AT-rich sequences, and consequently forming DNA secondary structure becomes a prominent cause to induced genome instability and CFS breakage. We added in new data and showed that depletion of FANCM increases instability not only at Flex1, but also at the AT rich sequences from FRA16C and FRA3B (new data in Fig.1d and data not shown). These data suggest that FANCM has a general role in protecting AT-rich sequences at CFSs.

Specific comments:

1. FANCD2 has been previously implicated in the protection of CFSs including FRA16D (Howlett et al. 2005 Human Mol Genet). Moreover, it has shown that FANCD2 is important for replication through CFSs including FRA16D (Madireddy et al. 2016 Mol Cell). BRCA1 is also well known to protect CFSs (Arlt et al. 2004 Mol Cell). It is very surprising that only FANCM but not other FA/BRCA proteins are required for suppression of recombination at the Flex1 reporter. The Flex1 reporter may not faithfully recapitulate the important features of CFSs.

Response: CFS expression (breakage) can be induced by various mechanisms. The HR-Flex reporter is designed to specifically monitor features associated with two important mechanisms (Figure 1 in this text). (1). One is the protection

Figure 1. HR-mediated DSB repair at Flex1 in the HR-Flex reporter.
 In the HR-Flex reporter, the EGFP open reading frame is disrupted by insertion of the Flex1 sequence. When DSBs are generated at Flex1, D-EGFP (donor EGFP) is used as a template to repair DSBs to restore the EGFP open reading frame to produce green cells. If the mechanisms to protect Flex1 is defective such as loss of FANCM, more DSBs would be generated causing increased HR to produce more green cells (top right box). However, if HR is defective such as loss of BRCA1, although more DSBs are accumulated at Flex1, HR cannot occur, so fewer green cells would be accumulated (bottom right box).

mechanism at AT-rich structure-forming DNA sequences. If a defect in protection results in DSB formation at Flex1 without influencing HR activity, more green signals would be generated from our reporter (Figure 1 in this text, right top box). For instance, FANCM deficiency results in more DSB formation but does not significantly influence HR, so we saw more green signals (Fig.1b, 1f and 2a). (2). The other is HR mechanism that is used to repair DSBs at Flex1. If HR is defective,

green signals would be reduced even if DSBs are formed at Flex1 (Figure 1 in this text, right bottom box). For instance, BRCA1 deficiency leads to impaired HR, thereby resulting in reduction of green cell accumulation (Supplementary Fig. 2a). Thus, our reporter assay suggests that FANCM plays a role in preventing DSB formation at Flex1, while BRCA1 is important for HR repair at Flex1. Both of these functions are important for maintaining CFS stability.

However, CFS expression can be caused by multiple mechanisms and those defects related to replication origin firing and transcription regulation would not be detected by our reporter as expected. FANCD2 is important for CFS protection including FRA16D, but the underlying mechanisms involve inhibition of dormant origin firing and prevention of DNA/RNA hybrid formation⁸. These features would not be detected in our HR-Flex reporter. Thus, although FANCD2, FANCA and possible other factors do not have a direct role in protecting Flex1 as revealed by our HR-Flex reporter, they are still important for CFS stability via other mechanisms. On the other hand, although HR-Flex reporter cannot score all defects causing CFS instability, it allows us to perform screening to identify new factors such as FANCM which are critical for maintaining CFS stability and to illustrate the underlying mechanisms.

2. In contrast to FRA16D, the fragility of FRA3B is thought to arise from low origin firing instead of DNA secondary structures and fork stalling (Letessier et al. 2011 Nature). It is very surprising that FANCM knockdown also increased the expression of FRA3B in Fig. 3.

Response: As described above, multiple mechanisms including paucity of replication origin, large-sized genes, fork stalling at unusual DNA sequences and late replication timing underlie CFS instability. The contribution of each factor to the fragility of a specific CFS may vary depending on the context of CFSs. For instance, certain CFSs may have fewer replication origins and also fewer structure-forming DNA sequences than others, and for these CFSs, low origin firing may be the predominant cause to induce CFS instability. On the other hand, some other CFSs may contain more AT rich sequences and thus formation of DNA secondary structures to stall DNA replication becomes a more important contributor for inducing instability of these CFSs.

We agree that a major mechanism to induce FRA3B instability under normal condition is low origin firing. However, FRA3B does contain AT-rich sequences prone to forming secondary structures. We cloned one such sequence and showed that this AT-rich sequence also induces mitotic recombination, and FANCM is important for the protection of this sequence (new data in Fig.1d and Supplementary Fig. 1c). Under normal conditions, the AT rich sequences at FRA3B are

protected by FANCM-mediated mechanism, while lack of origin may be the predominant mechanism causing FRA3B breakage. However, when FANCM is deficient, these AT rich sequences are not protected leading to significant more DSB accumulation and increased expression of FRA3B.

3. In Fig. 4, how Ras induces recombination at FRA16D and the Flex1 reporter is not explained at all. Oncogene activation could induce replication stress in many different ways, and most of these are very indirect. Without an understanding of how Ras induces CFS expression, it is difficult to appreciate the function of FANCM in this context.

Response: We provided new data to show that Ras expression induces replication stress as revealed by ATR checkpoint activation (Chk1 and RPA2 phosphorylation) and DSB formation (γ H2AX phosphorylation) (new data in Supplementary Fig. 9). We also performed ChIP analysis and showed that DSBs are accumulated at Flex1 upon Ras overexpression (new data in Fig. 4e). These data suggest that Ras induces replication stress which directly causes DSB formation at Flex1, thereby inducing recombination at Flex1 and causing CFS expression.

4. The results in Fig. 5a and 5b can be interpreted differently. Rad52 may be required for forming secondary structures, fork stalling, or DSB formation at the Flex1 reporter.

Response: In our model, we propose that when FANCM is deficient, DSBs accumulated at Flex1 would rely on Rad52 to repair. We provide new data to show that when Rad52 is depleted in FANCM deficient cells, DSB formation at Flex1 is increased as revealed by ChIP of γ H2AX (new data in Fig. 5d). This suggests that Rad52 is not required for DSB formation, but rather for suppression of DSB formation at Flex1 when FANCM is deficient. Thus, these data support our model that Rad52 is needed for repairing DSBs at Flex1 when FANCM function is impaired.

5. Based on the results in Fig. 5d, the authors proposed that Rad52 is important for HR when DSB ends are blocked by secondary structures. However, in the case of HR-Flex, DSB ends are blocked by both secondary structures and a non-homologous sequence. In the case of HR-Luc, DSB ends are blocked by a non-homologous sequence only. In neither case, DSB ends are blocked by secondary structures only. Therefore, the authors' hypothesis is not supported by experimental evidence.

Response: We performed new experiments by generating new reporters (HR-Flex/D-Flex and HR-Luc/D-Luc, Fig. 5f), which contain Flex1 or Luc sequences in the donor templates. In this way, the I-SceI cleavable EGFP receipt cassettes (EGFP::Flex1/I-SceI or EGFP::Luc/I-SceI) contain perfect homology to the donor templates (D-Flex or D-Luc), while Flex1 but not Luc would form secondary structures after end resection. Due to insertion of Flex1 or Luc in the donor templates, we cannot use green signals for detection of HR products. Instead, we used a PCR-based analysis. We introduced BamHI and EcoRI Sites to the donor templates, and if HR is used, the BamHI and EcoRI sites would be transferred to the receipt cassettes. Thus, BamHI/EcoRI cleavable PCR products among all (uncleavable bands are the products of imperfect end joining) would reflect HR efficiency (see more detailed description in the manuscript). As shown in Fig. 5f, inactivation of Rad52 significantly reduces HR products from Flex1-containing reporter, but not Luc-containing reporter. These results support the hypothesis that even when perfect homology is present at the donor templates, Rad52 is required for promoting HR at the DSB ends that are blocked by DNA secondary structures.

6. The interpretation of the results in Fig. 6 may not be correct. FANCM is important for the response to replication stress in various situations, and its function is clearly not limited to CFSs. Even if FANCM deficient cells are dependent on Rad52 for proliferation, it does not necessarily mean that the proliferation defect of the double mutant arises from problems at CFSs. The dependency of FANCM deficient cells on Rad52 is virtually the same as the previously reported Rad52 dependency of BRCA2 defective cells (Zeng et al. 2010 PNAS). This dependency could be explained by the redundant HR functions of FA/BRCA proteins and Rad52.

Response: We agree that the synthetic lethality phenotype of FANCM knockout and Rad52 knockdown is not necessarily caused only by the functions that we described at CFSs. We have a discussion on that in the “Discussion” section. In addition to CFSs, there are many other types of DNA secondary structures such as G quadruplexes (G4s). FANCM may play an important role to protect those sites in addition to CFSs, and in the absence of FANCM, Rad52 would also be needed for repair of DSBs at those sites as well. Thus, the proliferation defects of the double mutants could arise from problems more than CFSs.

However, the synthetic lethality interaction of FANCM and Rad52 is distinct from that of BRCA2 and Rad52 as reported previously, where it was proposed that Rad52 serves as an alternative

HR mechanism when HR is impaired in BRCA2 deficient cells⁹. FANCM is not essential for HR and its deficiency only causes mild impairment in HR, which is in sharp contrast to a strong HR defect observed in BRCA2 deficient cells. BRCA1 and BRCA2 knockout mice are embryonic lethal due to a HR defect, while FANCM mice grow normally. Loss of FANCM results in a substantial increase of HR-mediated mitotic recombination at Flex1, whereas inactivation of BRCA1 or BRCA2 leads to a reduction of HR at Flex1. Thus, the dependency of Rad52 and FANCM for cell survival is different from the redundant HR function of Rad52 for BRCA2.

Reviewer #2:

1. The part of Introduction describing the conclusion is too long and overlaps with Discussion. It should be shortened.

Response: We have revised the Introduction and simplified the conclusion part.

2. Figure 1c and 1d are out of the order in citation in the text. This should be corrected.

Response: We have corrected the order of Figures.

3. Page 7, 2nd paragraph, the authors should make clear to readers that the DNA binding activity of FAAP24 is required for recruitment of FAAP24 to CFS, whereas the FANCM interacting motif is dispensable for the same recruitment. As written, it is unclear to reader whether the authors are describing recruitment of FAAP24 or FANCM.

Response: We revised that part in the manuscript to make it clear that DNA binding of FAAP24 but not its interaction with FANCM is important for FAAP24 recruitment.

4. RAD52 and FANCM are synthetic lethal, suggesting that the two proteins work in two parallel pathways to promote cell proliferation. This synthetic lethal interaction is different from their

genetic interactions in CFS protection. The authors should make this clear to readers in the discussion.

Response: We propose that in the normal situation, FANCM pathway is used preferentially to protect Flex1 to prevent DNA secondary structure formation and subsequent DSB formation at Flex1, and thus Rad52 is minimally needed. However, when FANCM is deficient, massive DSBs are generated at Flex1, which then rely on Rad52 for repair. Thus, cells would die when Rad52 activity is inhibited in FANCM deficient cells. The synthetic lethality interaction of these two proteins is more likely due to their concerted and distinct roles, but not necessarily by paralleled activities. We added in more discussion to clarify our model based on identified mechanisms and synthetic lethality interaction of these two proteins.

5. Page 10, first paragraph, the authors stated that “these studies reveal an important function of RAD52 in mammalian cells in a special context when DNA ends are blocked by DNA secondary structure”. I did not see evidence for this statement. Figure 5d shows that the luciferase construct has the same effect as the Flex report. Do the authors think that the luciferase reporter can also form secondary DNA structures? If this statement is just speculation, please move this part to the Discussion.

Response: We have performed new experiments (Fig. 5f) and showed that even when perfect homology is present at the donor templates, Rad52 is still required for repairing DSBs at Flex1 but not Luc. This supports the model that Rad52 is required for HR when DSB ends are blocked by DNA secondary structures. Also see Response to Reviewer #1, comment 5.

Reviewer #3:

Major points:

1) Just one shRNA for each gene (FANCA, FANCD2, FAAP24, MHF1, RAD51, RAD52) was used for Figure 1b, 1c, Supplemental Figure 1c, Figure 5a,b,d. The authors should perform an "add-back experiment" by knocking down with shRNA followed by transfecting an expression plasmid to rescue the phenotype or use at least two shRNAs/gene. (For FANCM, the authors have performed the add-back experiments.)

Response: We have repeated several key experiments using two different shRNAs for each gene. New data have been added in Fig.1b, Supplementary Fig. 2, Supplementary Fig. 4, Supplementary Fig. 6 and Supplementary Fig.10a, 10b, 10c.

2) Usage of just one cancer cell line (U2OS or HCT116) for most of the experiments is a concern, since whether the conclusion can be generalized is questionable. The authors should use multiple cell lines for some key experiments.

Response: We have repeated key experiments using more cell lines. We showed that Flex1 also induces mitotic recombination in HeLa, MCF7 and T98G cells in addition to U2OS cells and inactivation of FANCM further increases genome instability at Flex1 as revealed by increased mitotic recombination (Fig. 1c and Supplementary Fig. 3a). In addition to U2OS cells, we also showed that Flex1 and other CFS-derived AT-rich sequences induce more mitotic recombination in FANCM KO HCT116 cells compared to wild type cells (new data in Fig. 1d and Supplementary Fig. 3b).

3) Figure 3e: the authors should test the effect of FANCM-MM1 mutant or depletion of other

FA proteins (FANCA, FANCD2, etc.) on fragile site breakage.

Response: The expression of CFSs in FANCA and FANCD2 deficient cells have been published¹⁰, which shows that loss of FANCA and FANCD2 leads to an increase of CFS breakage. We tested fragile site breakage at FRA16D in the FANCM-MM1 mutant and added in new data (Supplementary Fig. 8).

FANCM-MM1 mutant (defective in binding to the FA core complex) does not show significant defect in suppressing mitotic recombination at Flex1 (Fig. 1e), suggesting that the interaction of FANCM with the FA core complex is not important for Flex1 protection. However, FANCM-MM1 mutant shows an increased CFS expression (Supplementary Fig. 8), although not as significant as FANCM translocase mutant FANCM-K117R. This suggests that the interaction of FANCM with the FA core complex is still important for protection of CFSs, but mechanistically it is not through protecting the Flex1 site. In this regard, FANCA and FANCD2 are also important for CFS maintenance, but through other mechanisms. Consistently, loss of FANCA and FANCD2 activity does not cause hyper mitotic recombination at Flex1 (Fig. 1b, also see Response to Reviewer #1, specific comment 1).

4) Some statistical tests should be done for any quantitative data analyses.

Response: We have indicated statistical tests in figures.

Minor points:

1) Page 3 Line 21: “the FANCD2 and FANCI heterodimeric complex (ID2), which further recruits the downstream FA proteins including FANCD1/BRCA2, FANCI/BRIP1, FANCN/PALB2, and FANCO/RAD51 to DNA damage sites to promote HR” This sentence is incorrect. ID2 complex does not recruit BRCA2, BRIP1, PALB2 or RAD51C.

Response: Thanks to the reviewer for pointing out this. We have revised the text accordingly.

2) In reference 14 (Howlett, N.G., et al. Hum Mol Genet 14, 693-701 (2005)), it is reported that FANCD2 depletion in HCT116 leads to elevated fragile site instability following exposure to APH. The authors should comment on the interpretation of the discrepancy between ref 14 and the current paper.

Response: FANCD2 is important for CFS maintenance, but using mechanisms such as inhibition of dormant origin firing and prevention of DNA/RNA hybrid formation⁸, which are different from protection of AT-rich sequences at CFSs as described in this manuscript. Our HR reporter is specifically designed to detect HR-mediated repair of DSBs generated at Flex1 (see Figure 1 in this text), and it is expected that our reporter would not detect these mechanisms. Also see Response to Reviewer #1, specific comment 1.

References

1. Zhang, H. & Freudenreich, C.H. An AT-rich sequence in human common fragile site FRA16D causes fork stalling and chromosome breakage in *S. cerevisiae*. *Mol Cell* **27**, 367-79 (2007).
2. Wang, H. et al. CtIP maintains stability at common fragile sites and inverted repeats by end resection-independent endonuclease activity. *Mol Cell* **54**, 1012-21 (2014).
3. Ozeri-Galai, E. et al. Failure of origin activation in response to fork stalling leads to chromosomal instability at fragile sites. *Mol Cell* **43**, 122-31 (2011).

4. Ozeri-Galai, E., Bester, A.C. & Kerem, B. The complex basis underlying common fragile site instability in cancer. *Trends Genet* **28**, 295-302 (2012).
5. Mishmar, D. et al. Molecular characterization of a common fragile site (FRA7H) on human chromosome 7 by the cloning of a simian virus 40 integration site. *Proc Natl Acad Sci U S A* **95**, 8141-6 (1998).
6. Lukusa, T. & Fryns, J.P. Human chromosome fragility. *Biochim Biophys Acta* **1779**, 3-16 (2008).
7. Shah, S.N., Opresko, P.L., Meng, X., Lee, M.Y. & Eckert, K.A. DNA structure and the Werner protein modulate human DNA polymerase delta-dependent replication dynamics within the common fragile site FRA16D. *Nucleic Acids Res* **38**, 1149-62 (2010).
8. Madireddy, A. et al. FANCD2 Facilitates Replication through Common Fragile Sites. *Mol Cell* **64**, 388-404 (2016).
9. Feng, Z. et al. Rad52 inactivation is synthetically lethal with BRCA2 deficiency. *Proc Natl Acad Sci U S A* **108**, 686-91 (2011).
10. Howlett, N.G., Taniguchi, T., Durkin, S.G., D'Andrea, A.D. & Glover, T.W. The Fanconi anemia pathway is required for the DNA replication stress response and for the regulation of common fragile site stability. *Hum Mol Genet* **14**, 693-701 (2005).

Reviewers' Comments:

Reviewer #1:

Remarks to the Author:

The authors have performed a number of new experiments to improve the paper. However, some of the issues that I raised are not completely addressed.

General: I agree that many CFSs contain AT-rich structure-forming DNA sequences, but whether these sequences are sufficient for conferring fragility at endogenous CFSs is not always clear. Moreover, although the report assay developed by the authors is useful for characterizing specific AT-rich sequences derived from CFSs, whether the fragility observed in the reporter is the main source of fragility at endogenous CFSs is still unclear. For example, even if the AT-rich sequences from FRA3B are fragile in the reporter, are these sequences a significant source of fragility at endogenous FRA3B in cells? If the fragility of FRA3B comes from multiple sources, how significant is the contribution of these AT-rich sequences? It is also worth noting that all the AT-rich sequences tested in the reporter are from CFSs. What if some AT-rich sequences from non-fragile sites are tested in the reporter? It is clear that the reporter is a nice system to study the properties of AT-rich sequences, but whether the mechanistic details from this reporter are significantly relevant at endogenous CFSs is less clear.

1. I agree with the authors' explanation of why BRCA1 and FANCD2 did not display the same effects as FANCM in the reporter assay.

2. Are DSBs formed at the AT-rich sequences in FRA3B in FANCM knockdown cells? FANCM may have other functions in stabilizing forks in long genes.

3. How does Ras induce replication stress specifically at Flex1 (or CFSs)? If Ras induces replication stress globally, why are AT-rich sequences affected more than other regions of the genome?

4. The new data in Fig. 5d have addressed the concern.

5. I am confusing by the authors' interpretation. In Fig. 5d, HR-Luc is Rad52 dependent. In Fig. 5f, HR-Luc/D-Luc is Rad52 independent. If I combine these results, I would conclude that Rad52 is important for dealing with a non-homologous sequence at DSBs. If we compare HR-Flex/D-Flex and HR-Luc/D-Luc in Fig. 5f, Rad52 has a specific role in HR-Flex/D-Flex. However, it is not clear whether Rad52 is required at DNA end, donor, or both. At any rate, the data do not specifically support that Rad52 functions on secondary DNA structures at DNA ends.

6. I don't agree with the authors' argument. Although BRCA1 and BRCA2 deficient cells display stronger HR defects than FANCM mutant, it is impossible to exclude that FANCM Rad52 double mutant is not severely defective for HR. The authors should test this directly.

Reviewer #3:

Remarks to the Author:

The authors have addressed all the concerns I raised.

The manuscript has improved.

Minor points:

Figure 5f

The picture of one of the gels (the right one) is mislabeled; "HR-Flex/D-Flex" must be "HR-Luc/D-Luc".

Response to Reviewers' Comments

We thank the reviewers for the comments. In response to Reviewer 1's comments, we performed new experiments and have included additional new data in the manuscript (Fig. 3e and Supplementary Fig. 2b). We also corrected Figure 5 according to Reviewer 3's comment. Specific explanations are outlined below. The changes in the manuscript are marked by lines on the left side of the text.

Reviewer 1

General: I agree that many CFSs contain AT-rich structure-forming DNA sequences, but whether these sequences are sufficient for conferring fragility at endogenous CFSs is not always clear. Moreover, although the report assay developed by the authors is useful for characterizing specific AT-rich sequences derived from CFSs, whether the fragility observed in the reporter is the main source of fragility at endogenous CFSs is still unclear. For example, even if the AT-rich sequences from FRA3B are fragile in the reporter, are these sequences a significant source of fragility at endogenous FRA3B in cells? If the fragility of FRA3B comes from multiple sources, how significant is the contribution of these AT-rich sequences? It is also worth noting that all the AT-rich sequences tested in the reporter are from CFSs. What if some AT-rich sequences from non-fragile sites are tested in the reporter? It is clear that the reporter is a nice system to study the properties of AT-rich sequences, but whether the mechanistic details from this reporter are significantly relevant at endogenous CFSs is less clear.

Response: It has been shown that replication stalls at AT-rich sequences at endogenous CFSs¹, suggesting that AT-rich structure-forming DNA sequences contribute to fragility at endogenous CFSs. Our studies further showed that the expression of FRA16D and FRA3B is significantly increased when AT-rich sequences are not properly protected in FANCM deficient cells (Fig.3d). This suggests that AT-rich sequences at endogenous CFSs are indeed vulnerable sites for CFS breakage especially when the protection mechanism is defective.

As we discussed in the previous response, the presence of AT-rich sequences is one of the multiple mechanisms underlie CFS instability and the contribution of AT-rich sequences to fragility of a specific CFS may vary depending on the context of CFSs. AT-rich sequences at CFSs are normally protected by the FANCM pathway. The impact of these AT-rich sequences on CFS fragility becomes apparent when the protection mechanism is defective. For instance, we showed that endogenous FRA3B expression is increased in FANCM deficient cells (Fig.3d). We also presented new data that when FANCM is depleted, DSB formation is indeed increased around AT-rich sequences at the endogenous FRA3B locus (new data in Fig. 3e).

It is true that our reporter can also test the effect of AT-rich sequence from non-fragile sites as well as other structure-forming DNA sequences. In this aspect, it worth to note that many key regulators for keeping CFS stability also have general roles in protecting global genome stability. For instance, ATR has a general role in checkpoint activation and fork protection but identifying its specific role in CFS protection is still of great importance². Conversely, finding the role of certain proteins in CFS protection (e.g. maintaining stability of AT-rich sequences in CFSs) and then elucidating their more general roles (e.g. also important for maintaining stability of other structure prone DNA sequences) is also important. FANCM protects CFSs, but this does not exclude its function in protecting other places in the genome containing DNA secondary structures. We discussed the possible role of FANCM in protecting other structure-forming DNA sequences in the genome (p18 top). The relevance of the findings from our repair reporters to the mechanism of CFS maintenance has

been addressed by examining endogenous CFS expression (Fig. 3c, 3d and 3f) and DSB formation at endogenous CFSs including FRA3B (new data Fig. 3e).

1. I agree with the authors' explanation of why BRCA1 and FANCD2 did not display the same effects as FANCM in the reporter assay.

2. Are DSBs formed at the AT-rich sequences in FRA3B in FANCM knockdown cells? FANCM may have other functions in stabilizing forks in long genes.

Response: We performed new experiments using ChIP analysis of γ H2AX at AT-rich sequences in endogenous FRA3B locus and show that γ H2AX signals are significantly increased when FANCM is knocked down (new data in Fig.3e). This suggests that DSBs are accumulated around the AT-rich sequences at endogenous FRA3B in FANCM deficient cells and FANCM plays an important role in preventing DSB formation at the vicinity of AT rich sequences of FRA3B.

3. How does Ras induce replication stress specifically at Flex1 (or CFSs)? If Ras induces replication stress globally, why are AT-rich sequences affected more than other regions of the genome?

Response: Ras induces global replication stress, similar to HU or APH treatment. Due to replication stress, single stranded DNA (ssDNA) is accumulated at replication forks, which allows structure-prone DNA sequences (when they are in the ssDNA form), such as Flex1 and other AT-rich sequences, to form secondary structures at replication fork (see Figure 7, left and top, and see text). Thus, AT-rich sequences are affected more because they form DNA secondary structures upon replication stress, while other regions of the genome do not form secondary structures and thus would be affected less.

4. The new data in Fig. 5d have addressed the concern.

5. I am confusing by the authors' interpretation. In Fig. 5e, HR-Luc is Rad52 dependent. In Fig. 5f, HR-Luc/D-Luc is Rad52 independent. If I combine these results, I would conclude that Rad52 is important for dealing with a non-homologous sequence at DSBs. If we compare HR-Flex/D-Flex and HR-Luc/D-Luc in Fig. 5f, Rad52 has a specific role in HR-Flex/D-Flex. However, it is not clear whether Rad52 is required at DNA end, donor, or both. At any rate, the data do not specifically support that Rad52 functions on secondary DNA structures at DNA ends.

Response: Previously we showed in Fig. 5e, that Rad52 is required for both HR-Flex and HR-Luc. This suggests that Rad52 is required when DSB ends are blocked by a non-homologous sequence (in case of HR-Luc), but we cannot conclude whether DNA secondary structures can also block DSB ends leading to Rad52 dependence. We thus performed new experiments shown in Fig. 5f.

In Fig. 5f, we added in new drawing to illustrate repair intermediates of the EGFP::Flex1/I-SceI and EGFP::Luc/I-SceI substrates after I-SceI cleavage and end resection. In both HR-Flex/D-Flex and HR-Luc/D-Luc reporters, the I-SceI cleavage cassettes and the donors have perfect homology sequences. However, Flex1 but not Luc forms secondary structures at DSB ends after end resection. Our data showed that Rad52 is required for HR in HR-Flex/D-Flex but not HR-Luc/D-Luc, which suggests that secondary structures formed at DSB ends could also cause the reliance of HR on Rad52. We thus propose that when DSB ends are blocked (this can be either non-homologous sequences or DNA secondary structures), Rad52 is required for HR. Details of proposed mechanisms related to end blockage, strand invasion to the donors or second end capture, are illustrated in Supplementary Fig. 11.

6. I don't agree with the authors' argument. Although BRCA1 and BRCA2 deficient cells display stronger HR defects than FANCM mutant, it is impossible to exclude that FANCM Rad52 double mutant is not severely defective for HR. The authors should test this directly.

Response: We would like to emphasize that FANCM and BRCA1/BRCA2 play very different roles in HR and Flex1-induced mitotic recombination. FANCM has very minor defect in HR while BRCA1 and BRCA2 are essential for HR. Flex-induce mitotic recombination is substantially increased in FANCM deficient cells but significantly reduced in BRCA1 or BRCA2 deficient cells (Fig. 1b, 1c and 1d, and Supplementary Fig. 2a and 2b). We also have data indicating that HR remains at substantial levels when FANCM and Rad52 are both inactivated by shRNAs (Figure, see below). This further supports that the synthetic lethality of FANCM and Rad52 is not due to complete loss of HR as the cause for synthetic lethality of BRCA1/2 and Rad52 deficiency.

Figure. HR is maintained at substantial levels when both FANCM and Rad52 are deficient. U2OS (EGFP-HR) reporter cell line was infected with lentiviral shRNAs for FANCM shRNA#1, Rad52, or both, and I-SceI-induced HR was assayed. Knockdown efficiency was shown by Western blot analysis.

Reviewer #3

The authors have addressed all the concerns I raised.
The manuscript has improved.

Minor points:

Figure 5f

The picture of one of the gels (the right one) is mislabeled; “HR-Flex/D-Flex” must be “HR-Luc/D-Luc”.

Response: Thank the reviewer for pointing this out. We have corrected it.

References

1. Ozeri-Galai, E. et al. Failure of origin activation in response to fork stalling leads to chromosomal instability at fragile sites. *Mol Cell* **43**, 122-31 (2011).
2. Casper, A.M., Nghiem, P., Arlt, M.F. & Glover, T.W. ATR regulates fragile site stability. *Cell* **111**, 779-89 (2002).

Reviewers' Comments:

Reviewer #1:

Remarks to the Author:

The authors have addressed some but not all of my concerns.

In the new Fig. 3e, the control locus should be another site in FRA3B away from the AT-rich region, but not GAPDH. The question here is whether the fragility of FRA3B specifically arises from the AT-rich locus, but not to compare the fragility of FRA3B with a non-fragile site.

The I-SceI based HR assay using shFANCM and siRAD52 may be misleading. The HR at collapsed replication forks may not be identical to the HR at I-SceI breaks. FANCM can certainly be more important at collapsed forks.

I am still not satisfied by the authors' explanation of the "dirty end" model for Rad52. The model in Sup Fig. 11 involves cleavage of the non-homologous sequences at resected ends. However, there is no explanation of how Rad52 contributes to this event. A cleavage step may not be needed at the HR-Flex/D-Flex reporter, but it is hard to tell whether Rad52 functions at the DNA end or the donor Flex sequence.

Although I think that this manuscript still has some weaknesses, I appreciate the efforts that authors have made to improve it. I support the acceptance of this manuscript for publication.

Response to the comments from Reviewer #1:

1. In the new Fig. 3e, the control locus should be another site in FRA3B away from the AT-rich region, but not GAPDH. The question here is whether the fragility of FRA3B specifically arises from the AT-rich locus, but not to compare the fragility of FRA3B with a non-fragile site.

Response: When a DSB is generated, H2AX is phosphorylated and such phosphorylation is spread over several megabases in mammalian cells (1). Thus, it is impractical to use a site close to AT-rich region in FRA3B as a control locus for CHIP of γ H2AX. We are working on a linker-mediated ligation protocol to map DSB sites in CFSs, but technically it is very challenging when dealing with endogenous locus with low levels of DSBs and we hope to demonstrate that in another future publication.

2. The I-SceI based HR assay using shFANCM and siRAD52 may be misleading. The HR at collapsed replication forks may not be identical to the HR at I-SceI breaks. FANCM can certainly be more important at collapsed forks.

Response: The original question for this point (last round review question #6 from reviewer 1) is whether the FANCM and Rad52 synthetic lethality interaction is the same as that of BRCA1/2 with Rad52, which is caused by impaired HR when both are deficient. To address this point, I-SceI-based HR assay needs to be used to show whether HR is defective after DSB formation. When BRCA1/BRCA2 and Rad52 are both deficient, drastic reduction of HR frequency was observed after I-SceI cleavage (2,3). However, when we used the same repair assay, HR reduction is not obviously observed when both FANCM and Rad52 are depleted (see the figure in the response to #6 comment in the last round). Thus, the synthetic lethality of FANCM and Rad52 is not due to HR deficiency, which is distinct from that of BRCA1/2 and Rad52.

Whether HR is the same at collapsed forks and at DSBs on non-fork DNA is an important but different question. We have been working on generating reporters specifically monitoring DSB repair on collapsed forks, but that study is beyond the scope of this manuscript.

3. I am still not satisfied by the authors' explanation of the "dirty end" model for Rad52. The model in Sup Fig. 11 involves cleavage of the non-homologous sequences at resected ends. However, there is no explanation of how Rad52 contributes to this event. A cleavage step may not be needed at the HR-Flex/D-Flex reporter, but it is hard to tell whether Rad52 functions at the DNA end or the donor Flex sequence.

Response: In the discussion, we have proposed two possible mechanisms for the requirement of Rad52 to deal with HR when non-homologous sequences are present at DSB ends (p17). One is that Rad52 may be required for strand invasion when the 3' end invades the template with a blocked end. Alternatively, Rad52 may be required for second-end capture of the blocked end. To clarify the exact mechanism, extensive biochemical analysis and *in vivo* physical monitoring of step-by-step repair process will be needed in the future.